# Minimum Distance Summaries for Robust Neural Posterior Estimation

**Sherman Khoo** [1]   **Dennis Prangle** [1]   **Song Liu** [1]   **Mark Beaumont** [2]

## Abstract

Simulation-based inference (SBI) enables amortized Bayesian inference by first training a neural posterior estimator (NPE) on prior-simulator pairs, typically through low-dimensional summary statistics, which can then be cheaply reused for fast inference by querying it on new test observations. Because NPE is estimated under the training data distribution, it is susceptible to misspecification when observations deviate from the training distribution. Many robust SBI approaches address this by modifying NPE training or introducing error models, coupling robustness to the inference network and compromising amortization and modularity. We introduce minimum-distance summaries, a post-hoc robust NPE method that adapts queried test-time summaries independently of the pretrained NPE. Leveraging the maximum mean discrepancy (MMD) as a distance between observed data and a summary-conditional predictive distribution, the adapted summary displays strong robustness properties due to the robustness of the MMD. We demonstrate that the algorithm can be implemented efficiently with random Fourier feature approximations, yielding a lightweight, model-free test-time adaptation procedure. We provide theoretical guarantees for the robustness of our algorithm and empirically evaluate it on a range of synthetic and real-world tasks, demonstrating substantial robustness gains compared with existing robust SBI methods with minimal additional overhead.

## 1. Introduction

Across many areas of science and engineering, including cosmology (Alsing et al., 2019), biology (Beaumont et al., 2002), and neuroscience (Lueckmann et al., 2017), stochas-

---
[1]School of Mathematics, University of Bristol, Bristol, UK [2]Department of Biology, University of Bristol, Bristol, UK. Correspondence to: Sherman Khoo <sherman.khoo@bristol.ac.uk>.

*Proceedings of the 43rd International Conference on Machine Learning*, Seoul, South Korea. PMLR 306, 2026. Copyright 2026 by the author(s).

tic simulators are widely used as mechanistic statistical models. Such simulators are able to encode rich domain knowledge, and generate synthetic data for any candidate parameter, but their complexity makes statistical inference particularly challenging, as while simulating from the model is straightforward, the induced likelihood function $p(\mathbf{x} \mid \theta)$ is typically intractable or prohibitively expensive to evaluate, making standard likelihood-based inference challenging (Marin et al., 2012). This problem setting is known as likelihood-free inference, or simulation-based inference (SBI) (Cranmer et al., 2020).

Modern SBI methods are built around an *amortized* Bayesian inference paradigm, which is centered on training a conditional neural density estimator using simulated pairs $(\theta_i, \mathbf{x}_i)$ generated from the simulator model $p(\mathbf{x} \mid \theta)$ and the prior distribution $\pi(\theta)$ (Cranmer et al., 2020; Deistler et al., 2025). While these estimators can target the likelihood (Papamakarios et al., 2019), or density ratio (Cranmer et al., 2015; Hermans et al., 2020) $r(\mathbf{x}, \theta) = p(\mathbf{x} \mid \theta)/p(\mathbf{x})$, we focus here on the canonical neural posterior estimator (NPE) (Papamakarios & Murray, 2016), which directly learns a conditional density $q_\psi(\theta \mid \mathbf{x}) \approx p(\theta \mid \mathbf{x})$, thus mapping each observation, or typically, a summary statistic $\mathbf{s}$ to an approximation of its posterior distribution. After an initial, computationally heavy, offline training stage for the density estimator, the resulting estimator enables fast test-time statistical inference with a cheap forward pass. Test-time inference refers to post-training evaluations of the learned NPE $q_\psi(\theta \mid \mathbf{x})$ on new observed datasets, without further retraining of the estimated NPE, thus amortizing computational costs across future downstream queries. This paradigm is particularly attractive for many scientific workflows where inference is repeated across many downstream datasets and experimental conditions.

Nonetheless, because amortized estimators are trained under the simulator's prior predictive distribution $\mathrm{m}(\mathbf{x}) = \int p(\mathbf{x} \mid \theta)\pi(\theta)d\theta$, they are brittle under model misspecification. In SBI, model misspecification arises when the true DGP lies outside of the simulator family (Cannon et al., 2022), operationally, this is reflected in the mismatch between the prior-predictive distribution and the DGP (Schmitt et al., 2024; Elsemüller et al., 2025). Even a modest deviation from the true data-generating process (DGP) may lead to biased or unreliable posteriors (Ward et al., 2022; Schmitt

et al., 2024). Hence, there has been a growing literature of robust SBI approaches aimed at addressing this problem by reducing or accounting for this distributional discrepancy (Elsemüller et al., 2025; Kelly et al., 2025).

Although advances in robust SBI approaches have undoubtedly strengthened the real-world applicability of SBI, they often inherently couple robustness to the amortized inference method, for instance, by requiring observations during NPE training (Huang et al., 2023), modifying the training objective of the amortized estimator (Gao et al., 2023), or augmenting the simulator with additional modeling components (Ward et al., 2022). This coupling reduces modularity and undermines the central benefit of amortization: reusing a single pretrained NPE for cheap, repeated inference across downstream queries.

In this work, we focus on the classical Huber's contamination model as the form of misspecification, where the observed data distribution is contaminated with some fraction of arbitrary outlier distribution, and we address model misspecification by reducing the gap between the simulator-induced distribution and true DGP, while adhering to the design principle of ensuring robustness without compromising amortization. Concretely, we keep the pretrained NPE $q_\psi(\theta \mid \mathbf{s})$ fixed while modifying only the query of the NPE, i.e., the summary statistic of the observations. At test time, we adapt the observed data summary statistic $\mathbf{s}$ such that it minimizes a robust discrepancy between an estimated summary-conditional data distribution and the empirical distribution of observations. We call the resulting adapted summary statistic a *minimum-distance summary*, and perform robust inference by querying the fixed amortized NPE with this adapted summary. Consequently, this approach leads to a post-hoc process of robustifying a pre-trained NPE, which integrates smoothly into realistic Bayesian data analysis workflows (Gelman et al., 2020). This modular design aligns with the emerging trend towards reusable, general-purpose amortized SBI inference approaches (Vetter et al., 2026; Gloeckler et al., 2024), where amortized models are intended to be reused across a broad set of tasks, making a post-hoc, test-time robust method particularly desirable.

Our robust SBI approach aligns the data distribution and observations through a robust divergence, which we select as the maximum mean discrepancy (MMD) with a bounded characteristic kernel (Gretton et al., 2012), providing a robust, model-free objective for addressing misspecification. In order to provide a fast and lightweight estimation of the summary-conditional data distribution, we use a random Fourier feature approximation (Rahimi & Recht, 2007) which reduces the estimation to a regression problem, and yields a lightweight post-hoc robust method that is fully decoupled from the amortized inference approach. Additionally, since misspecification is rarely known a priori, we use the same MMD objective as a misspecification sensitive diagnostic, only invoking test-time adaptation when misspecification is detected.

## 2. Background

**Notation** Let $\theta \in \Theta \subseteq \mathbb{R}^{d_\theta}$ denote the model parameters, with a prior density $\pi(\theta)$. We denote a single observation as $\mathbf{x} \in \mathcal{X} \subseteq \mathbb{R}^{d_\mathbf{x}}$, and a dataset of $N$ independent and identically distributed (i.i.d.) observations as $\mathbf{x}_{1:N} = \{\mathbf{x}_n\}_{n=1}^N \in \mathcal{X}^N$. Let $\mathcal{P}(\cdot)$ represent the space of Borel measures on a given sample space. The simulator model $\mathbb{P}_\theta \in \mathcal{P}(\mathcal{X})$ defines an implicit likelihood $p(\mathbf{x} \mid \theta)$ for a single observation, and for a dataset of $N$ i.i.d. observations we have the joint distribution $p(\mathbf{x}_{1:N} \mid \theta) = \prod_{n=1}^N p(\mathbf{x}_n \mid \theta)$. We denote observed (possibly corrupted) datasets at test-time as $\tilde{\mathbf{x}}_{1:N}$ and the corresponding data-generating distribution as $\tilde{\mathbb{P}}$. For convenience, we provide a table of key expressions in our manuscript, which will be defined in subsequent sections.

| Symbol | Definition |
|---|---|
| $\mathbf{s}$ | Summary statistic of a dataset |
| $f(\mathbf{x}_{1:N})$ | Summary statistic function |
| $\mathbb{P}_{\mathbf{x}\mid\mathbf{s}}$ | Summary-conditional data distribution |
| $\hat{\mathbb{P}}_N$ | Empirical distribution of observed dataset $\tilde{\mathbf{x}}_{1:N}$ |
| $q_\psi(\theta \mid \mathbf{s})$ | Neural posterior estimator |
| $q_\omega(\mathbf{x} \mid \mathbf{s})$ | Decoder model |

### 2.1. Simulation-Based Inference

Simulation-based inference tackles Bayesian inference for implicit models $\mathbb{P}_\theta$ where the likelihood $p(\mathbf{x} \mid \theta)$ is not tractable but simulations $\mathbf{x} \sim \mathbb{P}_\theta$ can be drawn for any choice of $\theta \in \Theta$ (Cranmer et al., 2015). SBI methods estimate an *amortized* inference network using simulated parameter-data training samples, such that after this initial offline training stage, inference can be done during test-time for new observations cheaply with a single forward pass through the network. In this work, we focus on the canonical neural posterior estimator (NPE) (Papamakarios & Murray, 2016), which learns $q_\psi(\theta \mid \mathbf{x}) \approx p(\theta \mid \mathbf{x})$, where $\psi$ are parameters of a neural conditional density estimator. In high-dimensional data spaces, such as when we have a dataset of $N$ i.i.d. observations $\mathbf{x}_{1:N}$, we typically leverage a summary statistic $f : \mathcal{X}^N \to \mathcal{S}$ and perform inference on the summary space $q_\psi(\theta \mid \mathbf{s})$ where $\mathbf{s} = f(\mathbf{x}_{1:N})$. This summary statistic can either be hand-crafted (Fearnhead & Prangle, 2012), or learned with a neural encoder (Chen et al., 2021). Concretely, NPE uses training samples $\{(\theta^{(i)}, \mathbf{x}_{1:N}^{(i)}, \mathbf{s}^{(i)})\}_{i=1}^M$, $\mathbf{s} = f(\mathbf{x}_{1:N})$ that are drawn from the following generative process: $\theta \sim \pi(\theta)$, $\mathbf{x}_{1:N} \sim p(\mathbf{x}_{1:N} \mid \theta)$ and $\mathbf{s} = f(\mathbf{x}_{1:N})$. Then, the NPE is trained by minimizing the negative log-likelihood loss

$\mathcal{L}(\omega) = -\frac{1}{M} \sum_{i=1}^{M} \log q_\psi(\theta^{(i)} \mid \mathbf{s}^{(i)})$.

Thus, this procedure enables amortized inference, where after the initial NPE training stage, the NPE can be cheaply evaluated on any new observations at test-time. Beyond amortizing inference across observations, there has been a recent trend towards general-purpose inference approach, where pretrained models are reused across a wide range of inference tasks and setups. These include all-in-one models representing different conditional distributions of the joint simulator distribution (Gloeckler et al., 2024), training-free approaches (Vetter et al., 2026) leveraging probabilistic foundation models (Müller et al., 2025), and test-time procedures adapting amortized models to new priors (Yang et al., 2026).

## 2.2. Robust Simulation-Based Inference

Model misspecification arises when the data-generating distribution $\tilde{\mathbb{P}}$ lies outside of the simulator model family $\{\mathbb{P}_\theta : \theta \in \Theta\}$. In SBI, this is a particularly acute issue due to the additional model approximation from neural density estimation, and even a moderate mismatch has been shown to affect the accuracy and calibration of resulting posteriors (Ward et al., 2022; Cannon et al., 2022). Because the NPE is estimated from training samples, the distribution of training samples plays a key role in determining the effects of model misspecification, which is the prior-predictive distribution $\mathrm{m}(\mathbf{x}_{1:N}) = \int \left( \prod_{n=1}^{N} p(\mathbf{x}_n \mid \theta) \right) \pi(\theta) \, d\theta$.

This dependence can be made more explicitly by considering the negative log-likelihood objective function of the NPE, which can be shown to be equivalent to (up to additive constants) the expected KL-divergence between the true posterior and the NPE, under the prior-predictive distribution (Schmitt et al., 2024). Thus, when the observed data distribution deviates from the prior-predictive distribution, the quality of the NPE deteriorates. The difference between the DGP and the prior-predictive distribution, $\mathcal{D}[\mathrm{m}(\mathbf{x}_{1:N}), \tilde{\mathbb{P}}(\mathbf{x}_{1:N})]$ is thus known as the simulation gap (Schmitt et al., 2024).

There has been a growing literature of robust SBI approaches that have been developed in order to tackle model misspecification. Following the taxonomy in Kelly et al. (2025), existing approaches can be broadly grouped into three classes: (i) *robust summary approaches* that construct summary statistics that remove aspects of the data responsible for misspecification (Huang et al., 2023), (ii) *generalized Bayesian methods* that modify the likelihood in the standard Bayesian update rule with a robust loss or divergence (Bissiri et al., 2016; Dellaporta et al., 2022) and (iii) *error modeling* which introduces an explicit error model to account for the gap between simulations and observed data (Ratmann et al., 2009; Frazier & Drovandi, 2021). Although these approaches are effective, many methods inherently couple robustness with the inference procedure, resulting in robust inference procedures at the cost of amortization, such as using observed data during training of the NPE (Huang et al., 2023; Elsemüller et al., 2025) or additional error models (Ward et al., 2022; Kelly et al., 2024). Also their theoretical support is limited. However, concurrently with our work, Bharti et al. (2026) proposed a generalized Bayesian approach which is amortized and provably robust.

## 2.3. Maximum Mean Discrepancy-Based Inference

The maximum mean discrepancy (MMD) (Gretton et al., 2012) is a kernel-based discrepancy between two distributions $\mathbb{P}, \mathbb{Q}$ on $\mathcal{X}$, defined with respect to a positive definite kernel $k : \mathcal{X} \times \mathcal{X} \rightarrow \mathbb{R}$ and its reproducing kernel Hilbert space (RKHS) $\mathcal{H}$. Let $\phi : \mathcal{X} \rightarrow \mathcal{H}$ be a feature map such that $k(\mathbf{x}, \mathbf{y}) = \langle \phi(\mathbf{x}), \phi(\mathbf{y}) \rangle_\mathcal{H}$. The kernel mean embedding of $\mathbb{P}$ is $\mu_\mathbb{P} = \mathbb{E}_{\mathbf{x} \sim \mathbb{P}}[k(\mathbf{x}, \cdot)] = \mathbb{E}[\phi(\mathbf{x})]$ (Muandet et al., 2017), and the MMD is defined based on the distance between mean embeddings in the RKHS: $\mathrm{MMD}(\mathbb{P}, \mathbb{Q}) = \|\mu_\mathbb{P} - \mu_\mathbb{Q}\|_\mathcal{H}$. Motivated by the computational complexity of the MMD, random Fourier features (Rahimi & Recht, 2007) are an approximation to shift-invariant kernels using a finite-dimensional feature map $\mathbf{z} : \mathcal{X} \rightarrow \mathbb{R}^D$, allowing us to approximate the MMD as $\mathrm{MMD}(\mathbb{P}, \mathbb{Q}) \approx \|\mathbb{E}_{\mathbf{x} \sim \mathbb{P}}[\mathbf{z}(\mathbf{x})] - \mathbb{E}_{\mathbf{x} \sim \mathbb{Q}}[\mathbf{z}(\mathbf{x})]\|_2$. Crucially for the goal of tackling model misspecification, when the kernel is bounded, the MMD objective can be shown to have robustness properties under contamination (Chérief-Abdellatif & Alquier, 2022).

The MMD can be estimated directly from samples of the distributions $\mathbb{P}, \mathbb{Q}$, without requiring density evaluation, making the MMD a naturally well-suited divergence for simulator models. In particular, the MMD has been used for statistical inference of simulator models under the minimum distance estimation framework (Wolfowitz, 1957), and the resulting estimators have been shown to be consistent and robust under misspecification (Briol et al., 2019). MMD-based discrepancies have also been leveraged in approximate Bayesian computation (Park et al., 2016), generalized Bayesian inference (Cherief-Abdellatif & Alquier, 2020; Dellaporta et al., 2022), and more broadly in generative modeling (Li et al., 2015; Dziugaite et al., 2015).

## 3. Methodology

Recall that our goal is to develop a lightweight test-time procedure for robust SBI, that is fully decoupled from the pretrained neural posterior estimator (NPE). We introduce *minimum-distance summaries* (MDS), which, given test-time observations $\tilde{\mathbf{x}}_{1:N}$, selects an adapted summary $\mathbf{s}^*$ by minimizing the simulation-gap in *data-space*, specifically, we minimize a robust divergence between the summary-conditional data distribution $\mathbb{P}_{\mathbf{x}|\mathbf{s}}$ and the empirical distri-

bution of $\tilde{\mathbf{x}}_{1:N}$. Performing this alignment on data-space leverages robustness directly where misspecification is most interpretable, and crucially, because the NPE depends on observations only through the summary statistic query $\mathbf{s}$, the MDS $\mathbf{s}^*$ can be estimated and used directly in the NPE $q_\psi$.

## 3.1. Minimum Distance Summaries

Recall that the NPE $q_\psi(\theta \mid \mathbf{s})$ is trained under the prior-predictive distribution, which is generated from the following generative process, $\theta \sim \pi(\theta)$, $\mathbf{x}_{1:N} \mid \theta \sim \prod_{n=1}^{N} p(\mathbf{x}_n \mid \theta)$, and $\mathbf{s} = f(\mathbf{x}_{1:N})$. As our goal is to minimize the discrepancy in data-space, in order to connect a candidate summary statistic $\mathbf{s}$ to the data distribution, we consider the summary-conditional data distribution $\mathbb{P}_{\mathbf{x}|\mathbf{s}} = \mathrm{Law}(\mathbf{x}_1 \mid S = \mathbf{s}) \in \mathcal{P}(\mathcal{X})$. In particular, note that when $\mathbf{x}_{1:N}$ are i.i.d. samples, and $f$ is a permutation-invariant summary statistic, the conditional distribution $\mathbf{x}_{1:N} \mid S = \mathbf{s}$ is exchangeable over $N$ and consequently, conditional distributions $\mathrm{Law}(\mathbf{x}_n \mid S = \mathbf{s})$ are identical for all $n$.

Given a target distribution $\mathbb{Q} \in \mathcal{P}(\mathcal{X})$, we define the minimum-distance summary (MDS) as follows:

$$\mathbf{s}^\star(\mathbb{Q}) \;=\; \arg\min_{\mathbf{s} \in \mathcal{S}} \mathcal{D}\big(\mathbb{P}_{\mathbf{x}|\mathbf{s}}, \, \mathbb{Q}\big), \tag{1}$$

where $\mathcal{D}$ is a statistical divergence between the distributions defined on $\mathcal{X}$. For an observed dataset $\tilde{\mathbf{x}}_{1:N}$, we take $\mathbb{Q}$ to be the empirical distribution of the observed dataset, $\hat{\mathbb{P}}_N := \frac{1}{N} \sum_{n=1}^{N} \delta_{\tilde{\mathbf{x}}_n}$. The MDS objective can thus be seen as looking for the projection of the empirical distribution on the family of models $\{\mathbb{P}_{\mathbf{x}|\mathbf{s}} : \mathbf{s} \in \mathcal{S}\}$, analogously to minimum-distance estimation (Wolfowitz, 1957). Intuitively, the MDS $\mathbf{s}^*$ represents the summary statistic that best matches the empirical distribution, under a certain choice of divergence. As we will see in subsequent sections, the choice of divergence is crucial in ensuring robustness and computational efficiency of the MDS. We note here that the MDS objective is defined on *marginal* datapoints $\mathbf{x}_n$, instead of over the entire dataset $\mathbf{x}_{1:N}$, as we only observe a single realization of the observation dataset $\tilde{\mathbf{x}}_{1:N}$, making divergence estimation on this space statistically inefficient.

**Amortized Decoder Estimation** To compute the divergence $\mathcal{D}$ in Equation (1), we need the conditional distribution $\mathbb{P}_{\mathbf{x}|\mathbf{s}}$ given a particular $\mathbf{s}$. We approximate this distribution with an amortized (over summary statistics $\mathbf{s}$) conditional density estimator $q_\omega(\mathbf{x} \mid \mathbf{s})$, trained offline using the same training samples used for NPE training, $\{(\mathbf{x}_{1:N}^{(i)}, \mathbf{s}^{(i)})\}_{i=1}^{M}$ (allowing us to avoid additional simulations), which we call a *decoder model*, as it inverts the summary statistic back onto marginal data points.

Depending on the type of divergence $\mathcal{D}$, we may require exact densities, or possibly just samples from the distribution.

A flexible choice of neural density estimator would thus be a flow-based model (Papamakarios et al., 2017; Durkan et al., 2019), which allows for both exact density evaluation and sampling. A flow-based conditional density estimator can be used as a decoder model $q_\omega$ by training with the negative log-likelihood loss, $\mathcal{L}(\omega) = -\frac{1}{MN} \sum_{i=1}^{M} \sum_{n=1}^{N} \log q_\omega(\mathbf{x}_n^{(i)} \mid \mathbf{s}^{(i)})$.

Given the trained, amortized decoder model, during test-time, given observations $\tilde{\mathbf{x}}_{1:N}$, we can estimate the objective in Equation (1) with $\mathcal{D}(q_\omega(\mathbf{x} \mid \mathbf{s}), \hat{\mathbb{P}}_N)$, which can be optimized directly with respect to $\mathbf{s}$ to obtain the MDS. A standard choice of divergence would be the forward KL divergence (Kullback & Leibler, 1951), which is equivalent to likelihood maximization. A computational issue faced with using the KL divergence is that estimating the decoder model requires conditional density estimation, which is in general expensive and limits the general applicability of our method. As we shall see in the next section, using the MMD bypasses the need for full density estimation, thus allowing us to mitigate this computational challenge. Furthermore and crucially, the KL divergence objective is not considered robust to contamination as the log-likelihood is unbounded and a small fraction of outliers, with small model density can dominate the objective. Since our goal is to obtain a *robust* SBI approach, we require a robust divergence. A natural choice for a robust divergence would be the maximum-mean discrepancy (MMD). We provide a comparison of the MMD with the forward KL divergence in Appendix B.

## 3.2. Maximum Mean Discrepancy MDS

The maximum mean discrepancy (MMD) (Gretton et al., 2012) is a highly attractive choice of discrepancy $\mathcal{D}$ for MDS estimation as it has strong theoretical guarantees, in particular being robust under outlier contamination for bounded kernels (Chérief-Abdellatif & Alquier, 2022), and, as it depends on distributions only through their kernel mean embeddings, allows us to sidestep density estimation. Inheriting the robustness properties from the MMD, we provide theoretical results on the MMD based MDS in Section 4.

Recall that for two probability distributions $\mathbb{P}, \mathbb{Q} \in \mathcal{P}(\mathcal{X})$, the MMD is defined $\mathrm{MMD}^2(\mathbb{P}, \mathbb{Q}) = \|\mu_\mathbb{P} - \mu_\mathbb{Q}\|_\mathcal{H}^2$ where $\phi$ is the feature map associated with $k$, $\mathcal{H}$ is the corresponding RKHS, and $\mu_P := \mathbb{E}_{\mathbf{x} \sim \mathbb{P}}[\phi(\mathbf{x})]$ is the kernel mean embedding. Using this in our MDS objective in Equation 1,

$$\mathbf{s}^\star \;=\; \arg\min_{\mathbf{s} \in \mathcal{S}} \mathrm{MMD}^2\big(\mathbb{P}_{\mathbf{x}|\mathbf{s}}, \, \hat{\mathbb{P}}_N\big)$$

$$= \arg\min_{\mathbf{s} \in \mathcal{S}} \left\| \mu(\mathbf{s}) - \frac{1}{N} \sum_{n=1}^{N} \phi(\tilde{\mathbf{x}}_n) \right\|_\mathcal{H}^2,$$

where we have $\mu(\mathbf{s}) := \mathbb{E}[\phi(\mathbf{x}) \mid S = \mathbf{s}]$ as the summary-

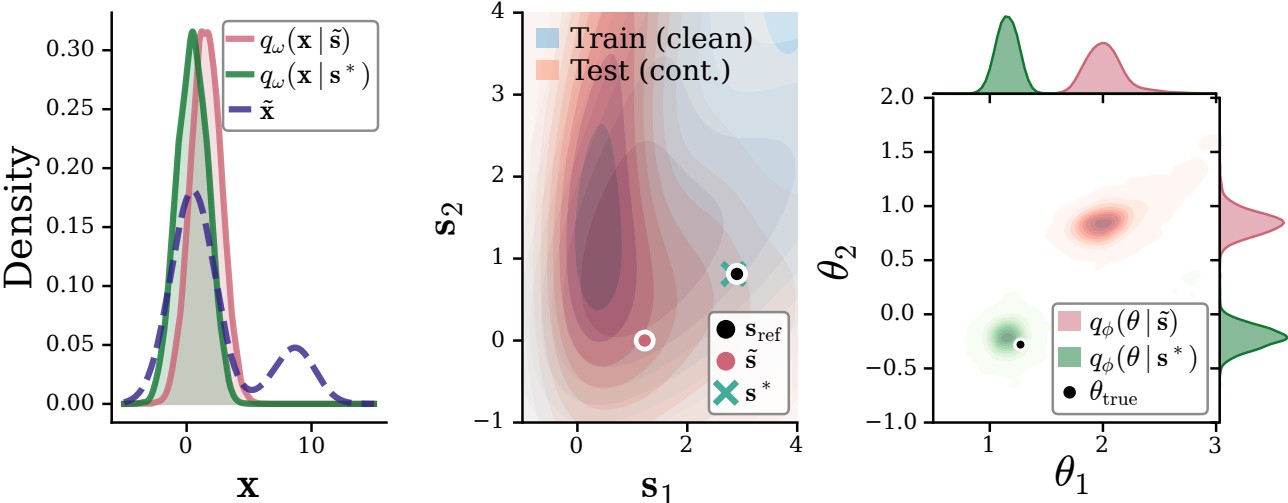

*Figure 1.* MDS for a bivariate Gaussian model with 20% of observations $\tilde{\mathbf{x}}_{1:100}$ contaminated by a shift of 8 units. $\mathbf{s}_{\text{ref}}, \tilde{\mathbf{s}}, \mathbf{s}^*$ are the oracle (uncontaminated) summary, test observation (contaminated) summary and adapted MDS summary respectively. Left: MDS aligns the decoder model $q_\omega$ with the contaminated observations $\tilde{\mathbf{x}}$ using a robust divergence. Middle: MDS is able to recover the oracle summary despite aligning on the data space. Right: MDS robustifies the NPE and the NPE is now able to recover the true parameters.

conditional mean embedding.

Crucially, note that the objective depends on the distribution $\mathbb{P}_{\mathbf{x}|\mathbf{s}}$ only through its conditional mean embedding $\mu(\mathbf{s})$. Conditional mean embeddings can be estimated directly (Song et al., 2009; 2013), but require computationally expensive Gram matrix inversions, which is especially problematic if the number of training samples is large.

With the goal of obtaining a fast, lightweight, test-time adaptation procedure, we leverage random Fourier features (Rahimi & Recht, 2007), which allows us to approximate the Gaussian kernel feature map with a finite dimensional feature map $\mathbf{z}(\cdot) \in \mathbb{R}^K$ based on the Fourier transform of the kernel. Under this approximation, the MDS objective can be further approximated as the Euclidean distance between finite-dimensional mean embeddings,

$$\text{MMD}^2\big(\mathbb{P}_{\mathbf{x}|\mathbf{s}}, \hat{\mathbb{P}}_N\big) \approx \left\| \mathbb{E}[\mathbf{z}(\mathbf{x}) \mid S = \mathbf{s}] - \frac{1}{N} \sum_{n=1}^{N} \mathbf{z}(\tilde{\mathbf{x}}_n) \right\|_2^2 \tag{2}$$

**Amortized Decoder Mean Embedding** By leveraging the random Fourier feature objective in Equation (2), we can now directly estimate the conditional mean embedding via regression. Specifically, using training samples (from the NPE training) $\{(\bar{\mathbf{z}}^{(i)}, \mathbf{s}^{(i)})\}_{i=1}^{M}$, where $\bar{\mathbf{z}}^{(i)} = \frac{1}{N} \sum_{n=1}^{N} \mathbf{z}(\mathbf{x}_n^{(i)})$ is the empirical mean embedding, we can estimate the conditional decoder mean embedding $\mu_\omega : \mathbb{R}^{d_{\mathbf{s}}} \to \mathbb{R}^K$, by simply minimizing a mean-squared error loss function, $\mathcal{L}(\omega) = \frac{1}{M} \sum_{i=1}^{M} \|\mu_\omega(\mathbf{s}^{(i)}) - \bar{\mathbf{z}}^{(i)}\|_2^2$. In practice, we parameterize the decoder mean embedding

$\hat{\mu}_\omega$ as a neural network, which is particularly suitable for predicting high dimensional random Fourier feature embeddings. In particular, note that, as with the density-based decoder model proposed earlier, the estimated decoder mean embedding $\hat{\mu}(\mathbf{s})$ is amortized with respect to $\mathbf{s}$, and can be train in a completely offline setting, without test observations.

**Test-Time MDS Adaptation** At test-time, given an observed dataset $\tilde{\mathbf{x}}_{1:N}$ and the estimated amortized decoder mean embedding $\hat{\mu}_\omega$, we can directly optimize the MDS objective in Equation (2) with $\mathbf{s}^* = \arg\min_{\mathbf{s}\in\mathcal{S}} \|\hat{\mu}_\omega(\mathbf{s}) - \frac{1}{N} \sum_{n=1}^{N} \mathbf{z}(\tilde{\mathbf{x}}_n)\|_2^2$. Compared to the density-based decoder model optimization, this objective is completely deterministic, and allows for the use of fast gradient-based optimization algorithms, e.g., quasi-Newton methods or line search. We leverage the L-BFGS algorithm (Liu & Nocedal, 1989), initialized at the observed summary statistic $\tilde{\mathbf{s}} = f(\tilde{\mathbf{x}}_{1:N})$. The resulting adapted summary is then used directly as an input to the pretrained NPE, providing a lightweight, robust, test-time procedure that is fully decoupled from the NPE. We provide the algorithm using stochastic gradient descent (SGD) in Algorithm 1, and an illustration of the MDS with the MMD in Figure 1. Furthermore, in many realistic modeling scenarios, misspecification is not known a-priori. Since our MDS method adapts the query summary statistics starting from the original observed summary, when combined with a calibration based heuristic, we propose a joint procedure that proceeds with the MDS adaptation only when model misspecification is detected, which we discuss in Appendix C.

**Algorithm 1** MMD minimum distance summary with SGD

1: Offline (Training-time)
2: **Inputs:** Training samples $\{(\mathbf{x}_{1:N}^{(i)}, \mathbf{s}^{(i)})\}_{i=1}^{M}$, decoder mean embedding model $\mu_\omega(\mathbf{s})$
3: Compute empirical mean embeddings for training samples $\bar{\mathbf{z}}^{(i)} = \frac{1}{N}\sum_{n=1}^{N} \mathbf{z}(\mathbf{x}_n^{(i)})$
4: Estimate decoder mean embedding $\hat{\mu}_\omega(\mathbf{s})$ with MSE objective $\mathcal{L}(\omega) = \frac{1}{M}\sum_{i=1}^{M} \|\mu_\omega(\mathbf{s}^{(i)}) - \bar{\mathbf{z}}^{(i)}\|_2^2$
5: **Return:** Decoder mean embedding $\hat{\mu}_\omega(\mathbf{s})$ (*Amortized*)
6: Online (Test-time)
7: **Inputs:** Observations $\tilde{\mathbf{x}}_{1:N}$, pretrained NPE $q_\psi(\theta \mid \mathbf{s})$, SGD step size $\eta$ and iterations $T$
8: Compute empirical mean embeddings for observations $\hat{\mu}_{\mathrm{obs}} = \frac{1}{N}\sum_{i=1}^{N} \mathbf{z}(\tilde{\mathbf{x}}_N)$
9: Compute initial summary statistic $\mathbf{s}_0 \leftarrow f(\tilde{\mathbf{x}}_{1:N})$
10: **for** $t = 1$ **to** $T$ **do**
11: $\quad \mathbf{s}_{t+1} \leftarrow \mathbf{s}_t - \eta \cdot \nabla_{\mathbf{s}} \|\hat{\mu}_\omega(\mathbf{s}_t) - \hat{\mu}_{\mathrm{obs}}\|_2^2$
12: **end for**
13: $\mathbf{s}^* \leftarrow \mathbf{s}_T$
14: **Return:** NPE evaluated at MDS: $q_\psi(\theta \mid \mathbf{s}^*)$

## 4. Theory

In this section, we provide theoretical guarantees on the robustness and consistency of the posteriors produced with our proposed minimum-distance summaries (MDS) with the maximum-mean discrepancy (MMD).

### 4.1. Robustness

We analyze the sensitivity of the posterior distribution obtained by querying with the MDS, under contamination of the target data distribution. We denote $\{\mathbb{P}_{\theta|\mathbf{s}} : \mathbf{s} \in \mathcal{S}\}$ as the family of summary posterior distributions defined on $\mathcal{P}(\Theta)$, and $\{\mathbb{P}_{\mathbf{x}|\mathbf{s}} : \mathbf{s} \in \mathcal{S}\}$ as the family of summary-conditional data distributions defined on $\mathcal{P}(\mathcal{X})$. We assume both families admit regular conditional distributions, for all $\mathbf{s} \in \mathcal{S}$. We note that these conditionals may be exact, or approximations, in our proposed MDS approach they correspond to the pretrained NPE $q_\psi$ and decoder model $q_\omega$ respectively.

Recall that, given a target distribution $\mathbb{Q} \in \mathcal{P}(\mathcal{X})$, the population MMD based minimum-distance summary objective is:

$$\mathbf{s}^*(\mathbb{Q}) = \arg\min_{\mathbf{s} \in \mathcal{S}} \mathrm{MMD}(\mathbb{P}_{\mathbf{x}|\mathbf{s}}, \mathbb{Q}),$$

where the MMD is computed with respect to a fixed kernel $k$. We assume a minimizer exists, and refer to Briol et al. (2019) for conditions guaranteeing this.

In order to model outlier contamination, we consider Huber's classic $\epsilon$ contamination model. For any $\mathbf{y} \in \mathcal{X}$ and $\epsilon \in [0, 1]$, define the contaminated distribution $\mathbb{Q}_{\epsilon,\mathbf{y}} = (1 - \epsilon)\mathbb{Q} + \epsilon\delta_{\mathbf{y}}$. Theorem 4.1 shows that any small contam-

ination to the target distribution $\mathbb{Q}$ leads to a proportionally small change (as measured by the KL divergence) to the resulting posterior approximation with the MDS.

**Theorem 4.1.** *Consider any $\mathbb{Q} \in \mathcal{P}(\mathcal{X})$ and $\mathbf{y} \in \mathcal{X}$, and let $\mathbb{Q}_{\epsilon,\mathbf{y}} = (1 - \epsilon)\mathbb{Q} + \epsilon\delta_{\mathbf{y}}$, for $\epsilon \in [0, 1]$. Then, under the conditions of Appendix A.1:*

$$\sup_{\mathbf{y} \in \mathcal{X}} \frac{d}{d\epsilon} \mathrm{KL}[\mathbb{P}_{\theta|\mathbf{s}^*(\mathbb{Q})}, \mathbb{P}_{\theta|\mathbf{s}^*(\mathbb{Q}_{\epsilon,\mathbf{y}})}]\Big|_{\epsilon=0} < \infty. \quad (3)$$

*Proof.* See Appendix A.3. □

**Proof Sketch** Our proof is composed of three parts. Firstly, the results of Briol et al. (2019) show that the MDS is stable under contamination, i.e., $\mathbf{s}^*(\mathbb{Q})$ and $\mathbf{s}^*(\mathbb{Q}_{\epsilon,\mathbf{y}})$ are close. Secondly, we show this implies the resulting likelihoods (based on observing summaries) are close. Finally, this lets us leverage the results of Sprungk (2020) to show that the posteriors are close in KL.

The robustness property (3) considers infinitessimal contamination. This is a variation on the *Kullback Leibler posterior influence function* (see Bharti et al., 2026) which considers instead a single observation being contaminated. The result in Theorem 4.1 could likely be extended to other divergences considered in Sprungk (2020). However it would be a stronger result to establish *global posterior robustness* (see Bharti et al., 2026) which requires pointwise robustness of the posterior density.

### 4.2. Consistency

We provide a consistency result for the MMD-based minimum-distance summary, showing that, when the model is correctly specified, using the MDS does not affect posterior consistency. Intuitively, if the posterior distribution conditioned on the original summary statistic $\mathbf{s}_N$ concentrates on the true parameter $\theta_0$ as $N \to \infty$, then the posterior distribution conditioned on the MDS $\mathbf{s}^*$ concentrates in a similar way as well. Note that we use $\mathbf{s}_N$ instead of $\mathbf{s}$ to highlight the dependency on the number of samples in the summary statistic $\mathbf{s}_N = f_N(\mathbf{x}_{1:N})$.

We can now state our result, showing consistency under the original summaries $\mathbf{s}_N$ implies consistency under the minimum distance summaries $\mathbf{s}_N^*$.

**Theorem 4.2.** *Under the conditions of Appendix A.4, if $\mathbb{P}_{\theta|\mathbf{s}_N}$ converges weakly to $\delta_{\theta_0}$ for almost every sequence $(\mathbf{x}_i)_{i\geq 1}$ sampled under $\theta_0$, then $\mathbb{P}_{\theta|\mathbf{s}_N^*}$ converges weakly to $\delta_{\theta_0}$ as well.*

*Proof.* See Appendix A.5 □

**Proof Sketch** The proof shows that the predictive mixture distribution induced by $\mathbb{P}_{\theta|\mathbf{s}_N^*}$ converges to the true

data-generating distribution $\mathbb{P}_{\theta_0}$, and then uses a strong identifiability assumption to ensure $\mathbb{P}_{\theta|\mathbf{s}_N^*}$ converges to $\delta_{\theta_0}$.

# 5. Related Works

As discussed in Section 2.2, there is a growing body of robust SBI approaches, and MDS fits most naturally as a *robust summaries* method (Kelly et al., 2025), but differs in its focus as a lightweight, test-time robust method that preserves amortization. A related line of work is also motivated by minimizing the simulation-gap with a modified summary statistic as with MDS, but does so by introducing a regularization term during joint training of a NPE with a neural summary statistic network (Huang et al., 2023; Elsemüller et al., 2025), which can be framed as a form of unsupervised domain adaptation. In contrast, MDS minimizes the simulation-gap in data-space and preserves amortization by adapting purely on the summary queries. A complementary direction introduces explicit error models to account for the simulation-gap (Ward et al., 2022; Kelly et al., 2024). The robust NPE of Ward et al. (2022) augments NPE with a spike-and-slab error model, and performs inference by integrating over latent variables. While statistically principled, such approaches require additional modeling assumptions and nontrivial test-time computations (additional MCMC over latent variables). Our proposed MDS, by leveraging the MMD, is model-free, and has a lightweight test-time adaptation in the form of a deterministic optimization problem.

# 6. Numerical Experiments

In this section, we empirically evaluate our proposed minimum-distance summary with the MMD, focusing on the lightweight decoder mean embedding approach as discussed in Section 3.2. For all experiments, we adopt Huber's contamination model, where the observed test data $\tilde{\mathbf{x}}_{1:N}$ is drawn i.i.d. from $\tilde{\mathbb{P}} = \varepsilon\mathbb{Q} + (1-\varepsilon)\mathbb{P}_{\theta^*}$, where $\varepsilon \in \{0.0, 0.1, \ldots, 0.5\}$, $\theta^*$ is the ground-truth parameter and $\mathbb{Q}$ is a task-specific contamination. For all experiments presented in this section, we use $N = 100$ i.i.d. observations in the datasets. We focus on tasks with a fixed summary statistic, except for the Gaussian setup in Section 6.1, which uses a learned neural summary statistic.

Throughout our experiments, we maintain a standard NPE which is used as a non robust baseline. We implement the noisy NPE (NNPE) from Ward et al. (2022), which is based on a spike-and-slab error model, an alternate outlier removal baseline method, which pre-processes test observations with outlier detection, which we call NPE-OR, and finally, for the Gaussian setup, the NPE-RS of Huang et al. (2023); Elsemüller et al. (2025). Further details and additional results of all experiments are provided in Appendix F. We

use the misspecification detection calibration scheme as discussed in Appendix C, which we find improves the performance for the well-specified test examples, while still maintaining the robustness from adaptation during contamination. Further ablations with respect to the sample size and data dimension is provided in Appendix D. [1]

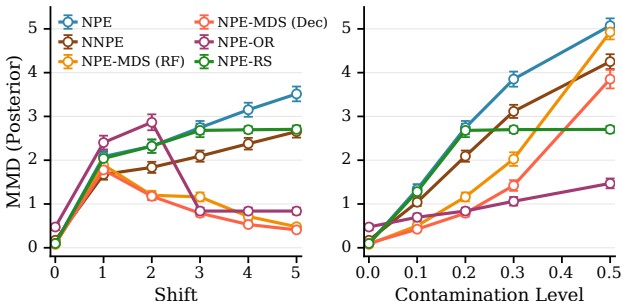

*Figure 2.* Bivariate Gaussian model with outlier contamination, comparing posterior samples against the groundtruth posterior distribution with MMD. Left: Increasing outlier magnitude. Right: Increasing contamination proportion.

## 6.1. Gaussian Model

We consider a conjugate bivariate Gaussian location model, with a Gaussian prior, which provides an analytic posterior distribution as a ground truth reference distribution. We induce contamination by replacing data points with outliers, with equal probability in both the positive and negative direction, with increasing magnitude of both the outlier points, and the proportion of contamination. We further implement both the MDS with the full decoder model [*MDS (Dec)*] and MDS with the random Fourier feature based decoder mean embedding [*MDS (RF)*]. From Figure 2, we can see that, in general, our MDS improves robustness compared to other benchmark methods, although for severe proportion of contamination, MDS can degrade. Theoretically, we expect the MDS to be guaranteed only for relatively small levels of contamination. Furthermore, note that on the left plot of Figure 2, we see that there is a sudden drop in NPE-MDS and NPE-OR methods despite increasing outlier shift. We attribute this to the misspecification detection procedure resulting in false negatives for smaller levels of outlier shift. We furthermore evaluate the same task on the NPE-PFN model (Vetter et al., 2026), and show that the MDS can provide test-time robustness in a similar fashion for foundation model-based SBI approaches, which is provided in Appendix E.

## 6.2. Time-Series Examples

We evaluate two time-series models with progressing levels of difficulty based on the dimensionality of the data-space.

---

[1]Our implementation is publicly available here

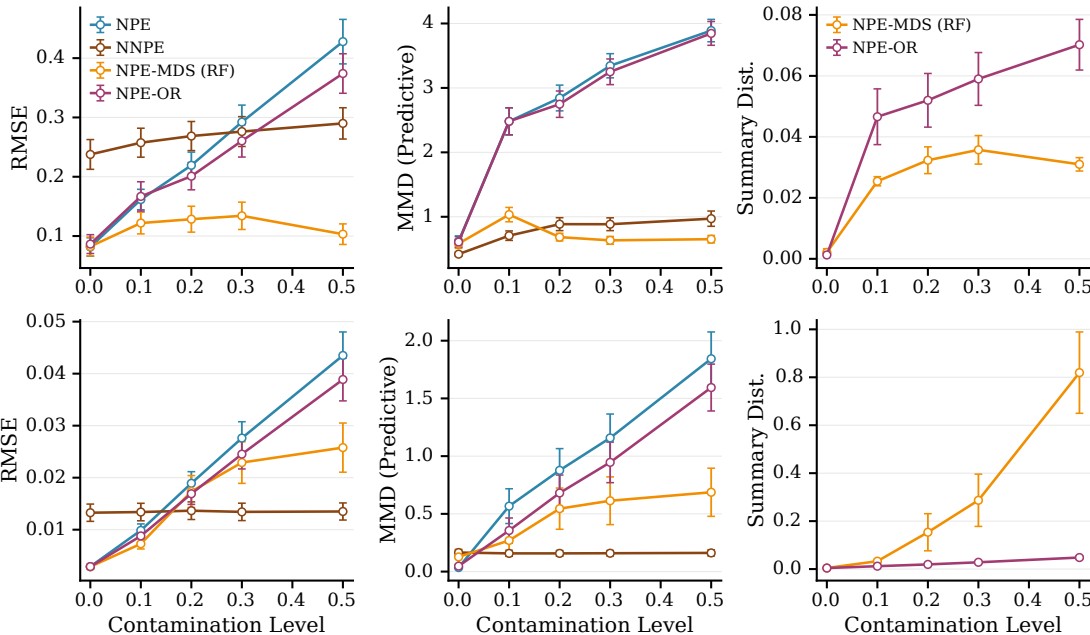

*Figure 3.* Time-series models with increasing contamination proportion, measuring the RMSE against the true parameter (**left**), posterior predictive against uncontaminated observations with MMD (**center**), and distance of adapted summary to the uncontaminated (oracle) summary statistic (**right**). Top: Ornstein-Uhlenbeck process. Bottom: SIR model.

**Ornstein-Uhlenbeck Process**   We consider an OUP with a two-dimensional parameter $\theta$ and trajectories of length $T = 25$, following a similar setup to Huang et al. (2023). The misspecification is induced by contaminating observation trajectories generated with the ground truth parameter $\theta^*$ with outlier samples generated from an OUP model with an alternate contamination parameter $\theta_c$. As shown in the top row of Figure 3, MDS-MMD yields consistently improved robustness across contamination levels. Notably, even though the MDS objective minimizes the simulation-gap in the data space, the adapted summary remains close to the clean, uncontaminated observation summary.

**SIR Model**   Next, we consider the SIR model from Ward et al. (2022), with a two-dimensional parameter space and trajectories of length $T = 365$. This follows the same Huber contamination model setup as described previously, but with a task-specific contamination distribution $\mathbb{Q}$, where we apply a weekend-reporting delay to the observed time series, providing a structural form of misspecification. With reference to the bottom row of Figure 3, MDS improves upon the standard NPE and NPE-OR, but does not match NNPE. Interestingly, even when the adapted summary deviates from the oracle summary under stronger misspecification, the degradation in posterior metrics is limited, which we attribute to partial non-identifiability of the summary statistic for this model. Additional experiments in Appendix F.3.1 show that the robustness benefits of MDS is retained even as the fraction of contamination increases up to $\varepsilon = 1.0$.

### 6.3. Cryo-EM Inference

Finally, we consider a realistic and challenging simulator model, based on cryo-electron microscopy (cryo-EM) (Dingeldein et al., 2025). Here, each observation is a $32 \times 32$ image ($d_{\mathbf{x}} = 1024$) obtained by projecting a 3D biomolecule under unknown orientation, shift, and measurement noise. The goal is to infer a one-dimensional shape index from the image. We induce misspecification by replacing an $\varepsilon$ fraction of images in the test observations with Gaussian noise (Figure 5). Despite the high dimensionality and the severe nature of the contamination, MDS is able to substantially improve the robustness of the NPE to contamination (Figure 4).

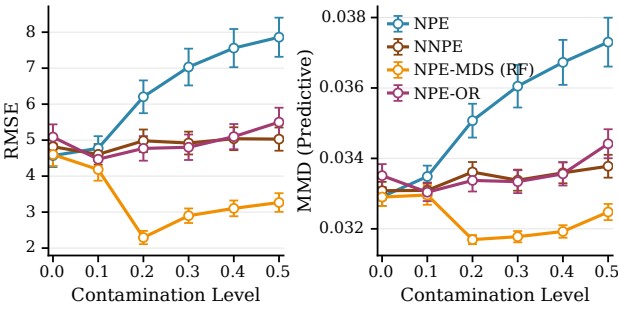

*Figure 4.* Cryo-EM inference. Left: RMSE against true parameter. Right: Posterior predictive against uncontaminated observations.

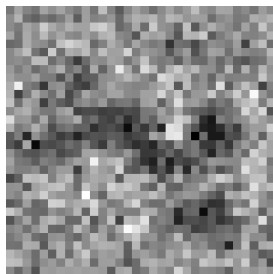 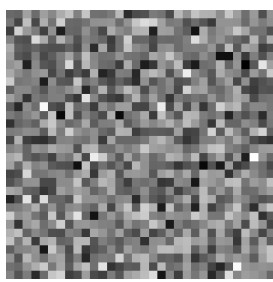

*Figure 5.* Left: Projected image with HSP90 model. Right: Gaussian noise contamination.

# 7. Conclusion

We introduced minimum-distance summaries (MDS), a lightweight, test-time robustness procedure for pretrained NPEs. MDS provides an adapted summary by minimizing a *robust* divergence between the data distribution and observations, mitigating misspecification, and crucially, obtains the robust adapted summary independently of the pretrained NPE. Thus, MDS is able to preserve the amortization property of the NPE, providing a modular approach that can be flexibly integrated in a broader statistical workflow. By leveraging the MMD as the divergence, we are able to provide a model-free, lightweight, and robust objective for the MDS. We establish theoretical guarantees of the MDS posterior consistency and stability under contamination. Across a range of synthetic and high-dimensional benchmarks, MDS improves robustness under contamination while maintaining strong performance in the well-specified regime.

**Limitations**   MDS is currently tailored towards a pretrained NPE utilizing summary statistics, and so does not apply to methods using the full data directly without any summary statistics. Since the MDS divergence operates on the data-space, for high-dimensional or structured data, specific choices of MMD kernels or learned feature representation may be necessary.

**Future Works**   While the MDS is a general framework for any divergence, we focused on the MMD, and an interesting future direction would be to explore problem settings motivating the use of alternate divergences, such as the Stein or Fisher divergence. Investigating the use of MDS for other amortized SBI methods, such as neural likelihood estimation, would also be useful to further broaden the applicability of MDS. Furthermore, extending the MDS to sequential NPE could be further considered. Naively, this would involve first estimating the MDS initially, then using the estimated MDS for subsequent sequential training. However, further methodological development could incorporate an adaptive MDS within the sequential NPE training.

## Acknowledgements

SK was supported by the EPSRC Center for Doctoral Training in Computational Statistics and Data Science, grant number EP/S023569/1. The authors thanks Sam Power, François-Xavier Briol and Charita Dellaporta for helpful discussions, as well as the three anonymous reviewers for their insightful comments.

## Impact Statement

This paper presents work whose goal is to advance the field of Machine Learning. There are many potential societal consequences of our work, none which we feel must be specifically highlighted here.

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

# A. Assumptions and Proofs for Theorems 4.1 and 4.2

The main paper uses the notation $\mathbb{P}_\theta$ for the distribution of $\mathbf{x}$ under a particular choice of $\theta$. In this appendix we instead write $\mathbb{P}_{\mathbf{x}|\theta}$. This is to make it more obvious which distribution this refers to, and to be consistent with notation for other conditional distributions.

## A.1. Assumptions for Theorem 4.1 (Robustness)

First we have assumptions to use results from Sprungk (2020).

1. $\mathbb{P}_{\theta|\mathbf{s}}$ has a Lebesgue density proportional to $\pi(\theta) \exp[-\Phi_\mathbf{s}(\theta)]$ where $\Phi_\mathbf{s} : \Theta \to \mathbb{R}$.

2. For any $\mathbf{s} \in \mathcal{S}$, $\mathrm{ess\,inf}_\mu \, \Phi_\mathbf{s} = 0$, where $\mu$ is the prior measure. Here, $\mathrm{ess\,inf}_\mu \, \Phi_\mathbf{s} := \sup\{a \in \mathbb{R} | \mu(\{\Phi_\mathbf{s}(\theta) < a\}) = 0\}$.

*Remark* A.1. Here $\Phi_\mathbf{s}(\theta)$ acts as a negative log-likelihood based on observing $\mathbf{s}$. Assumption 2 essentially requires that, for any $\mathbf{s}$, the corresponding likelihood is bounded above almost everywhere under the prior. Then $k(\mathbf{s}) := \mathrm{ess\,inf}_\mu \, \Phi_\mathbf{s}$ is the resulting greatest lower bound on $\Phi_\mathbf{s}$. So $\Phi'_\mathbf{s}(\theta) := \Phi_\mathbf{s}(\theta) - k(\mathbf{s})$ satisfies assumption 2 while giving the same $\mathbb{P}_{\theta|\mathbf{s}}$.

We introduce assumptions related to the sensitivity of the log-likelihood to $\mathbf{s}$.

3. For any $\theta \in \Theta$, $\Phi_\mathbf{s}(\theta)$ is a differentiable function in $\mathbf{s}$. Let $h(\mathbf{t}, \theta) = \nabla_\mathbf{s} \Phi_\mathbf{s}(\theta)|_{\mathbf{s}=\mathbf{t}}$.

4. For any compact $K \subseteq \mathcal{S}$ there exists $g_K(\theta)$ such that $\sup_{\mathbf{t} \in K} ||h(\mathbf{t}, \theta)|| \le g_K(\theta)$ and $\int g_K(\theta)\pi(\theta)d\theta < \infty$.

5. The summary statistic space $\mathcal{S} \subseteq \mathbb{R}^{d_\mathbf{s}}$ is a convex set.

Next we state regularity conditions on $\mathbb{P}_{\mathbf{x}|\mathbf{s}}$ to prove robustness of $\mathbf{s}^*(\mathbb{Q})$ (Lemma A.3 below).

6. The MMD kernel $k$ is bounded.

7. $\mathbb{P}_{\mathbf{x}|\mathbf{s}}$ has a Lebesgue density $p_\mathbf{s}(\mathbf{x})$.

8. For all $\mathbf{x}$, $p_\mathbf{s}(\mathbf{x}) \in C^2(\mathcal{S})$.

9. For all $i$, $\int \sup_{\mathbf{s} \in \mathcal{S}} \left| \frac{\partial}{\partial \mathbf{s}^i} p_\mathbf{s}(\mathbf{x}) \right| d\mathbf{x} < \infty$.

10. We consider $\mathbb{Q}$ such that $M(\mathbf{s}^*(\mathbb{Q}))$ is non-singular, where

$$M(\mathbf{t}) := \int H_\mathbf{s} \xi(\mathbf{y}, \mathbb{P}_{\mathbf{x}|\mathbf{s}})\Big|_{\mathbf{s}=\mathbf{t}} p_\mathbf{s}(\mathbf{y})d\mathbf{y},$$

$$\xi(\mathbf{y}, \mathbb{P}_{\mathbf{x}|\mathbf{s}}) := k(\mathbf{y}, \mathbf{y}) - 2\int k(\mathbf{y}, \mathbf{z})p_\mathbf{s}(\mathbf{z})d\mathbf{z} + \int k(\mathbf{z}, \mathbf{z}')p_\mathbf{s}(\mathbf{z})p_\mathbf{s}(\mathbf{z}')d\mathbf{z}d\mathbf{z}',$$

and $H_\mathbf{s}$ denotes the Hessian with respect to $\mathbf{s}$.

*Remark* A.2. Assumptions 7–9 are based on Key et al. (2025) (their assumption 3). Briol et al. (2019) give alternatives (their assumptions i–iii) which are instead based on a generative form of $\mathbb{P}_{\mathbf{x}|\mathbf{s}}$. Assumption 10 relates to a local identifiability condition (see Newey & McFadden, 1994, discussion under Theorem 3.1).

## A.2. Influence Function Lemma

Define the influence function for $\mathbf{s}^*$ as

$$\mathrm{IF}(\mathbf{y}; \mathbb{Q}) := \lim_{\epsilon \to 0} \frac{1}{\epsilon}[\mathbf{s}^*(\mathbb{Q}_{\epsilon, \mathbf{y}}) - \mathbf{s}^*(\mathbb{Q})].$$

Our main proof uses the following lemma, which is part of Theorem 5 from Briol et al. (2019). Since we use different assumptions, we provide a proof for completeness.

**Lemma A.3.** *Under assumptions 6–10,* $\sup_{\mathbf{y} \in \mathcal{X}} || \mathrm{IF}(\mathbf{y}; \mathbb{Q})|| < \infty$.

*Proof.* From Dawid & Musio (2014),

$$\text{IF}(\mathbf{y}; \mathbb{Q}) = M(\mathbf{s}^*(\mathbb{Q}))^{-1} \nabla_{\mathbf{s}} \xi(\mathbf{y}, \mathbb{P}_{\mathbf{x}|\mathbf{s}}) \Big|_{\mathbf{s}=\mathbf{s}^*(\mathbb{Q})}.$$

From assumption 10 it suffices to show that $\sup_{\mathbf{y}} ||\nabla_{\mathbf{s}} \xi(\mathbf{y}, \mathbb{P}_{\mathbf{x}|\mathbf{s}})|| < \infty$. To show this, first note that assumption 9 and the dominated convergence theorem allow interchange of differentiation and integration to give

$$\nabla_{\mathbf{s}} \xi(\mathbf{y}, \mathbb{P}_{\mathbf{x}|\mathbf{s}}) = -2 \int k(\mathbf{y}, \mathbf{z}) \nabla_{\mathbf{s}} p_{\mathbf{s}}(\mathbf{z}) d\mathbf{z} + 2 \int k(\mathbf{z}, \mathbf{z}') \nabla_{\mathbf{s}} p_{\mathbf{s}}(\mathbf{z}) p_{\mathbf{s}}(\mathbf{z}') d\mathbf{z} d\mathbf{z}'.$$

Applying the triangle inequality,

$$\left\| \nabla_{\mathbf{s}} \xi(\mathbf{y}, \mathbb{P}_{\mathbf{x}|\mathbf{s}}) \right\| \le 2 \left\| \int k(\mathbf{y}, \mathbf{z}) \nabla_{\mathbf{s}} p_{\mathbf{s}}(\mathbf{z}) d\mathbf{z} \right\| + 2 \left\| \int k(\mathbf{z}, \mathbf{z}') \nabla_{\mathbf{s}} p_{\mathbf{s}}(\mathbf{z}) p_{\mathbf{s}}(\mathbf{z}') d\mathbf{z} d\mathbf{z}' \right\|$$

$$\le 2 \sum_{i=1}^{d_{\mathbf{s}}} \left[ \int \left| k(\mathbf{y}, \mathbf{z}) \frac{\partial}{\partial \mathbf{s}^i} p_{\mathbf{s}}(\mathbf{z}) \right| d\mathbf{z} + \int \left| k(\mathbf{z}, \mathbf{z}') \frac{\partial}{\partial \mathbf{s}^i} p_{\mathbf{s}}(\mathbf{z}) p_{\mathbf{s}}(\mathbf{z}') \right| d\mathbf{z} d\mathbf{z}' \right].$$

Hence

$$\left\| \nabla_{\mathbf{s}} \xi(\mathbf{y}, \mathbb{P}_{\mathbf{x}|\mathbf{s}}) \right\| \le 4 \sup_{\mathbf{z}, \mathbf{z}'} k(\mathbf{z}, \mathbf{z}') \sum_{i=1}^{d_{\mathbf{s}}} \int \left| \frac{\partial}{\partial \mathbf{s}^i} p_{\mathbf{s}}(\mathbf{z}) \right| d\mathbf{z}.$$

From the assumptions, the right hand side is a finite bound for all $\mathbf{y}$, as required. $\square$

### A.3. Proof of Theorem 4.1 (Robustness)

For $\mathbf{s}, \mathbf{t} \in \mathcal{S}$, consider

$$||\Phi_{\mathbf{s}}(\theta) - \Phi_{\mathbf{t}}(\theta)||_{L_\mu^1} := \int |\Phi_{\mathbf{s}}(\theta) - \Phi_{\mathbf{t}}(\theta)| \pi(\theta) d\theta.$$

By the mean value theorem, $\Phi_{\mathbf{s}}(\theta) - \Phi_{\mathbf{t}}(\theta) = (\mathbf{s} - \mathbf{t})^T h(\mathbf{u}(\theta), \theta)$, where $\mathbf{u}(\theta) = a(\theta)\mathbf{s} + [1 - a(\theta)]\mathbf{t}$ for some $a : \Theta \to [0, 1]$. By assumption 5, $\mathbf{u}(\theta) \in \mathcal{S}$. Then

$$||\Phi_{\mathbf{s}}(\theta) - \Phi_{\mathbf{t}}(\theta)||_{L_\mu^1} \le ||\mathbf{s} - \mathbf{t}|| \int ||h(\mathbf{u}(\theta), \theta)|| \pi(\theta) d\theta. \tag{4}$$

Consider $\mathbf{t} \in K(\mathbf{s}) := \{\mathbf{t} : ||\mathbf{s} - \mathbf{t}|| \le 1\}$. By assumption 4, the integral in (4) is finite, so

$$||\Phi_{\mathbf{s}}(\theta) - \Phi_{\mathbf{t}}(\theta)||_{L_\mu^1} \le k_1(\mathbf{s})||\mathbf{s} - \mathbf{t}|| \tag{5}$$

for some $k_1 : \mathcal{S} \to \mathbb{R}$.

Under assumptions 1–2, we can apply Theorem 11 of Sprungk (2020)[2] to get

$$\text{KL}[\mathbb{P}_{\theta|\mathbf{s}}, \mathbb{P}_{\theta|\mathbf{t}}] \le k_2(\mathbf{s}) ||\Phi_{\mathbf{s}}(\theta) - \Phi_{\mathbf{t}}(\theta)||_{L_\mu^1} \tag{6}$$

for some $k_2 : \mathcal{S} \to \mathbb{R}$, when $||\Phi_{\mathbf{s}}(\theta) - \Phi_{\mathbf{t}}(\theta)||_{L_\mu^1} \le 1$.

Lemma A.3 gives:

$$\sup_{y \in \mathcal{X}} \lim_{\epsilon \to 0} \frac{1}{\epsilon} ||\mathbf{s}^*(\mathbb{Q}) - \mathbf{s}^*(\mathbb{Q}_{\epsilon, \mathbf{y}})|| < \infty. \tag{7}$$

For the remainder of the proof we fix $\mathbb{Q}$. From (7), there exists some $\eta : \mathcal{X} \to (0, \infty)$ such that $\epsilon < \eta(\mathbf{y})$ gives $||\mathbf{s}^*(\mathbb{Q}) - \mathbf{s}^*(\mathbb{Q}_{\epsilon, \mathbf{y}})|| \le 1/k_1(\mathbf{s}^*(\mathbb{Q}))$ and, by (5), $||\Phi_{\mathbf{s}^*(\mathbb{Q})}(\theta) - \Phi_{\mathbf{s}(\mathbb{Q}_{\epsilon, \mathbf{y}})}(\theta)||_{L_\mu^1} \le 1$. Then for $\epsilon < \eta(\mathbf{y})$,

$$\frac{1}{\epsilon} \text{KL}[\mathbb{P}_{\theta|\mathbf{s}^*(\mathbb{Q})}, \mathbb{P}_{\theta|\mathbf{s}^*(\mathbb{Q}_{\epsilon, \mathbf{y}})}] \le k_2(\mathbf{s}^*(\mathbb{Q})) \frac{1}{\epsilon} ||\Phi_{\mathbf{s}^*(\mathbb{Q})}(\theta) - \Phi_{\mathbf{s}^*(\mathbb{Q}_{\epsilon, \mathbf{y}})}(\theta)||_{L_\mu^1} \qquad \text{by (6)}$$

$$\le k_1(\mathbf{s}^*(\mathbb{Q})) k_2(\mathbf{s}^*(\mathbb{Q})) \frac{1}{\epsilon} ||\mathbf{s}^*(\mathbb{Q}) - \mathbf{s}^*(\mathbb{Q}_{\epsilon, \mathbf{y}})|| \qquad \text{by (5)}.$$

---

[2]The theorem gives (in our notation) a factor $k_2(\mathbf{s}, \mathbf{t})$. Equation (4) of Sprungk (2020) and the surrounding discussion shows how this translates into $k_2(\mathbf{s})$ for a suitable set of $\mathbf{t}$ values.

So for any $\mathbf{y} \in \mathcal{X}$,

$$\lim_{\epsilon \to 0} \frac{1}{\epsilon} \mathrm{KL}[\mathbb{P}_{\theta|\mathbf{s}^*(\mathbb{Q})}, \mathbb{P}_{\theta|\mathbf{s}^*(\mathbb{Q}_{\epsilon,\mathbf{y}})}] \leq k_1(\mathbf{s}^*(\mathbb{Q})) k_2(\mathbf{s}^*(\mathbb{Q})) \lim_{\epsilon \to 0} \frac{1}{\epsilon} ||\mathbf{s}^*(\mathbb{Q}) - \mathbf{s}^*(\mathbb{Q}_{\epsilon,\mathbf{y}})||.$$

Using (7) gives the required result,

$$\sup_{y \in \mathcal{X}} \lim_{\epsilon \to 0} \frac{1}{\epsilon} \mathrm{KL}[\mathbb{P}_{\theta|\mathbf{s}^*(\mathbb{Q})}, \mathbb{P}_{\theta|\mathbf{s}^*(\mathbb{Q}_{\epsilon,\mathbf{y}})}] < \infty.$$

### A.4. Assumptions for Theorem 4.2 (Consistency)

We assume:

1. The MMD kernel $k$ is bounded, continuous and integrally strictly positive definite (Simon-Gabriel et al., 2023).

2. $\mathbb{P}_{\mathbf{x}|\theta}$ is a pushforward measure $G_\theta^\# \mathcal{U}$ where $(\mathcal{U}, \mathcal{F}, \mathbb{U})$ is a probability measure and $G_\theta : \mathcal{U} \to \mathcal{X}$ is measurable.

3. For all $u \in \mathcal{U}$, $G_\theta(u)$ is a continuous function in $\theta$.

4. Consider the sequence $\mathbb{T}_N \in \mathcal{P}(\theta)$ and the resulting mixture distributions $\mathbb{Q}_N(A) = \int \mathbb{P}_{\mathbf{x}|\theta}(A) d\mathbb{T}_N$. If $\mathbb{Q}_N$ weakly converges to $\mathbb{P}_{\mathbf{x}|\theta_0}$, then $\mathbb{T}_N$ weakly converges to $\delta_{\theta_0}$.

*Remark* A.4. Assumption 1 ensures that MMD metrizes weak convergence (see Simon-Gabriel et al., 2023, Theorem 1). That is, $\mathbb{V}_N$ weakly converges to $\mathbb{V}$ if and only MMD$(\mathbb{V}_N, \mathbb{V}) \to 0$, where the sequence $\mathbb{V}_N$ and $\mathbb{V}$ are in $\mathcal{P}(\mathcal{X})$.

### A.5. Proof of Theorem 4.2 (Consistency)

We prove this in a particular setting. Consider a parametric family $\mathbb{P}_\theta \in \mathcal{P}(\mathcal{X})$ for $\theta \in \Theta$. Suppose that we have independent random variables $\mathbf{x}_i \sim \mathbb{P}_{\theta_0}$ for $i \in \mathbb{N}_0$ and some true parameter value $\theta_0 \in \Theta$. Given $N$, we treat $\mathbf{x}_{1:N}$ as observable, and $\mathbf{x}_0$ as a separate test variable. Let $\hat{\mathbb{P}}_N$ be the empirical distribution $\frac{1}{N} \sum_{i=1}^N \delta_{\mathbf{x}_i}$. Let $\mathbf{s}_N = f_N(\mathbf{x}_{1:N})$ where $f_N : \mathcal{X}^n \to \mathcal{S}$.

For all $N$, our setting gives regular conditional distributions $\mathbb{P}_{\theta|\mathbf{s}}^N \in \mathcal{P}(\Theta)$ (summary posterior), and $\mathbb{P}_{\mathbf{x}_0|\mathbf{s}}^N \in \mathcal{P}(\mathcal{X})$ (predictive decoder). These are the exact distributions given the random variables introduced above. This result does not consider approximation error.

The superscripts denote the dependence of these conditional distributions on $N$.

Define:

$$\mathbf{s}_N^* = \arg\min_{\mathbf{s} \in \mathcal{S}} \mathrm{MMD}(\mathbb{P}_{\mathbf{x}_0|\mathbf{s}}^N, \hat{\mathbb{P}}_N).$$

We assume that this $\arg\min$ exists (as before, see Briol et al. (2019) for conditions guaranteeing this).

We will need two weak convergence requirements[3]

$$\mathbb{P}_{\theta|\mathbf{s}_N} \to \delta_{\theta_0}, \tag{8}$$

$$\hat{\mathbb{P}}_N \to \mathbb{P}_{\mathbf{x}|\theta_0}. \tag{9}$$

From the theorem statement we will assume (8) hold for almost every sequence $(\mathbf{x}_i)_{i \geq 1}$ sampled under $\theta_0$. From Varadarajan's theorem (see Dudley, 2018, Theorem 11.4.1) the same is true for (9). So it is almost sure that both hold. For the remainder of the proof we fix $(\mathbf{x}_i)_{i \geq 1}$ such that this is the case.

**Part 1** We will show weak convergence of $\mathbb{P}_{\mathbf{x}|\mathbf{s}_N}^N$ to $\mathbb{P}_{\mathbf{x}|\theta_0}$.

Let $\theta_N$ be a random variable with distribution $\mathbb{P}_{\theta|\mathbf{s}_N}^N$. Select any continuous bounded $h : \mathbb{R}^{d_x} \to \mathbb{R}$. By assumption 3, (8) and the continuous mapping theorem, $\mathrm{plim}\, h(G_{\theta_N}(\mathbf{u})) = h(G_{\theta_0}(\mathbf{u}))$. Since $h$ is bounded we can apply dominated convergence to give

$$\mathbb{E}[h(G_{\theta_N}(\mathbf{u}))] \to \mathbb{E}[h(G_{\theta_0}(\mathbf{u}))].$$

Thus we get the required result

$$\mathop{\mathbb{E}}_{\mathbb{P}_{\mathbf{x}|\mathbf{s}_N}^N} [h(\mathbf{x})] \to \mathop{\mathbb{E}}_{\mathbb{P}_{\mathbf{x}|\theta_0}} [h(\mathbf{x})].$$

---

[3] Recall the notation $\mathbb{P}_{\mathbf{x}|\theta}$ was introdcued at the start of Appendix A.

**Part 2** We will show $\mathrm{MMD}(\mathbb{P}^N_{\mathbf{x}|\mathbf{s}_N}, \hat{\mathbb{P}}_N) \to 0$.

By the triangle inequality

$$\mathrm{MMD}(\mathbb{P}^N_{\mathbf{x}|\mathbf{s}_N}, \hat{\mathbb{P}}_N) \leq \mathrm{MMD}(\mathbb{P}^N_{\mathbf{x}|\mathbf{s}_N}, \mathbb{P}_{\mathbf{x}|\theta_0}) + \mathrm{MMD}(\mathbb{P}_{\mathbf{x}|\theta_0}, \hat{\mathbb{P}}_N).$$

Both terms converge to zero. For the first this follows from Part 1 and Remark A.4. For the second term this follows from (9) and Remark A.4.

**Part 3** We will show $\mathrm{MMD}(\mathbb{P}^N_{\mathbf{x}|\mathbf{s}^*_N}, \mathbb{P}_{\mathbf{x}|\theta_0}) \to 0$.

By the triangle inequality

$$\mathrm{MMD}(\mathbb{P}^N_{\mathbf{x}|\mathbf{s}^*_N}, \mathbb{P}_{\mathbf{x}|\theta_0}) \leq \mathrm{MMD}(\mathbb{P}^N_{\mathbf{x}|\mathbf{s}^*_N}, \hat{\mathbb{P}}_N) + \mathrm{MMD}(\mathbb{P}_{\mathbf{x}|\theta_0}, \hat{\mathbb{P}}_N)$$

Again, the second term converges to zero by (9) and Remark A.4. To show the first term converges to zero, define

$$A_N = \mathrm{MMD}(\mathbb{P}^N_{\mathbf{x}|\mathbf{s}^*_N}, \hat{\mathbb{P}}_N), \quad B_N = \mathrm{MMD}(\mathbb{P}^N_{\mathbf{x}|\mathbf{s}_N}, \hat{\mathbb{P}}_N).$$

Then $A_N \geq 0$, and $A_N \leq B_N$ from the definition of $\mathbf{s}^*_N$. Since $B_N \to 0$ by Part 2, we have $A_N \to 0$.

**Part 4** Finally, $\mathbb{P}^N_{\theta|\mathbf{s}^*_N}$ weakly converges to $\delta_{\theta_0}$ by Part 3, assumption 4 and Remark A.4.

## B. Choice of Divergence

As discussed in Section 3.1, the choice of divergence for the minimum-distance summaries objective, Equation (1) is crucial, as our main goal is to obtain a *robust* MDS. In order to do so, we have opted to use the MMD, which has strong robustness properties that is inherited with the MDS. Alternatively, when using the decoder model which estimates the summary-conditional data distribution with a flow-based model, such models allow for exact density evaluation, and we are able to use the forward KL divergence, which corresponds to maximum-likelihood.

We provide empirical results comparing the KL-divergence based MDS with the MMD MDS 3.2 in Figure 7, under the same Gaussian setup as Appendix F.1. Figure 6 shows the results for increasing contamination level $\varepsilon$, and Figure 7 shows the results for increasing contamination shift $\delta$. As we can see, this experiment verifies that the MMD is indeed robust, and that the KL is not a robust objective, as it performs similarly to the baseline NPE under contamination. Note that in Figure 7, we see that as the contamination shift increases, posterior RMSE improves. This is due to the behavior of the MMD, particularly the bounded Gaussian kernel. As the shift magnitude in the outliers increase relative to the inlier points, the influence of outliers is diminished, and so large-shift outliers affect the downstream accuracy of MDS less than moderately shifted outliers.

## C. Detecting Model Misspecification

The MMD was initially motivated as a two-sample test (Gretton et al., 2012) and has been exploited to detect model misspecification in previous SBI work (Schmitt et al., 2024). We propose a simple procedure to detect model misspecification using our trained decoder mean embedding $\hat{\mu}_\omega(\mathbf{s})$. We do this by calibrating a null distribution with a false positive rate $\alpha$, on held-out validation data pairs $(\mathbf{x}_{1:N}, \mathbf{s})$, specifically, by calculating the test statistics $\hat{\mathrm{MMD}} = \|\mu_\omega(\mathbf{s}) - \frac{1}{N}\sum_{n=1}^N \mathbf{z}(\mathbf{x}_n)\|_2^2$, and taking the $1 - \alpha$ quantile of this test statistics distribution, $\tau$. Then, we only proceed with the adaptation for an observed dataset $\tilde{\mathbf{x}}$ if $\|\hat{\mu}_\omega(\mathbf{s}_0) - \hat{\mu}_{\mathrm{obs}}\|_2^2 > \tau$, where $\mathbf{s}_0 = f(\tilde{\mathbf{x}})$.

We provide empirical results showing the empirical Type-I error and power of the model misspecification procedure in Figures 8 and 9 using the Gaussian setup in Appendix F.1. In particular, note that the left plot of Figure 9 confirms the sudden drop result discussed in Section 6.1, as this is due to the false negatives from the contamination magnitude not being large enough to trigger adaptation.

## D. Ablation Studies

In this section, we provide a series of ablation studies to test the robustness of our methods in a range of experimental settings.

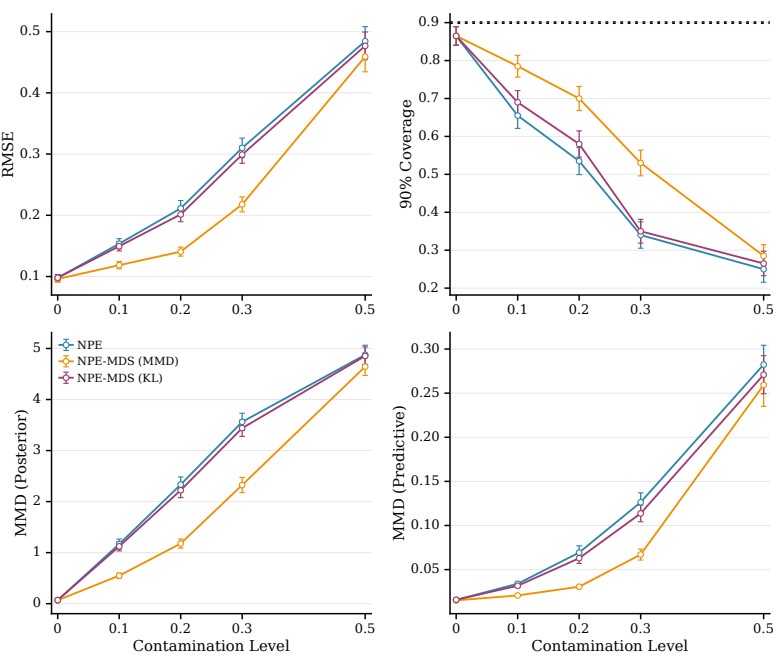

*Figure 6.* Comparison between KL-based MDS and MMD-based MDS, for increasing contamination levels.

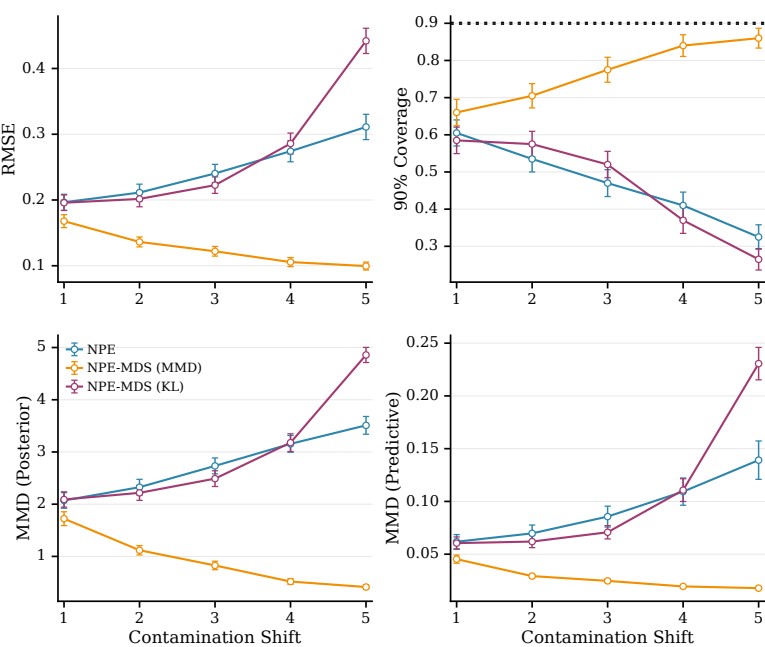

*Figure 7.* Comparison between KL-based MDS and MMD-based MDS, for increasing contamination shift.

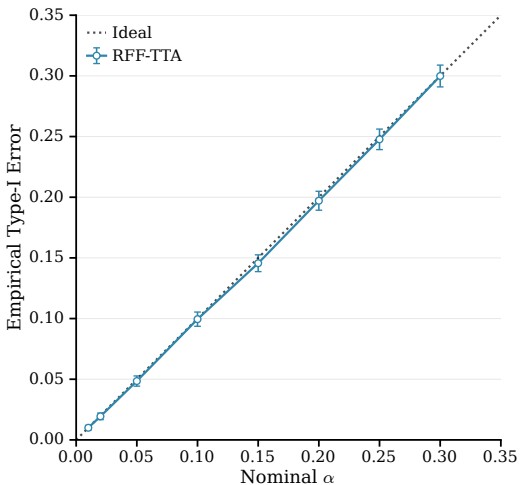

*Figure 8.* Empirical false positive rate for misspecification detection using the Gaussian task in Appendix F.1

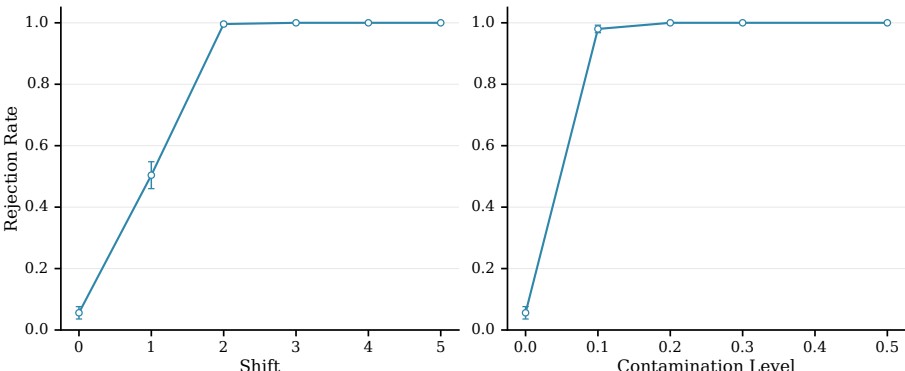

*Figure 9.* Rejection rate for misspecification detection using the Gaussian task in Appendix F.1, with increasing outlier shift and contamination proportion

For section D.1 and D.2, we test the sensitivity of our proposed MDS approach against increasing number of observations $N$ and observation dimension $d_{\mathbf{x}}$, under a Gaussian model setting as specified in Appendix F.1.

For section D.3 and D.4, we investigate the sensitivity of our optimization initialization and kernel bandwidth, using the OUP model as specified in Appendix F.2.

### D.1. Sample Size Ablation

We use broadly the same setup as in Appendix F.1, but changing the number of observations $N \in \{5, 10, 50, 100, 200, 500\}$. For the misspecification, we use $\delta = 5, \epsilon = 0.2$. We furthermore use directly the sample mean as the summary statistic. The results are provided in Figure 10. We find that overall, NPE-MDS obtains better robustness compared to both NPE and NNPE, although the improvement increases as $N$ increases.

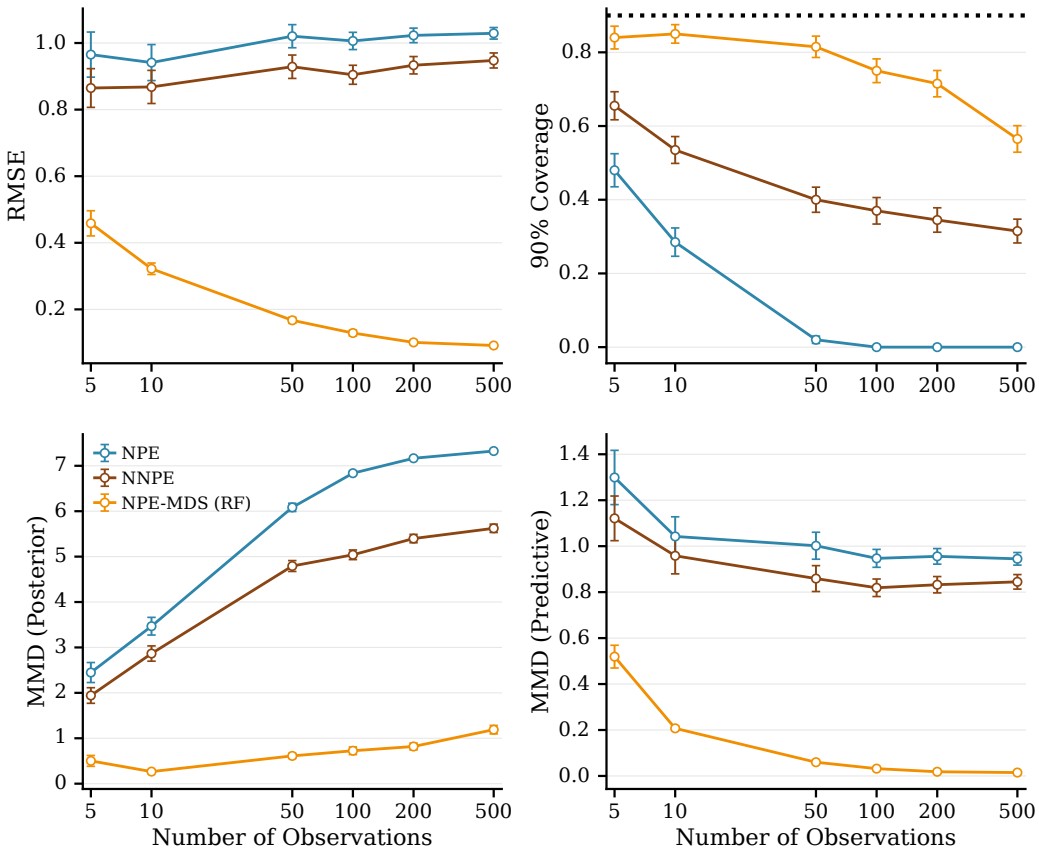

*Figure 10.* Results for ablation with increasing number of observations (Appendix D)

### D.2. Data Dimension Ablation

As we would like to investigate the sensitivity of our method to increasing observation dimension, increasing the dimensions of the Gaussian task directly would not be suitable as that would lead to an increase in the parameter and importantly, the summary statistic dimension. In order to keep both these variables fixed, we propose a slight alteration to the Gaussian task, using a Gaussian factor model $\mathbf{x}_i \mid \theta \sim N(A\theta, I)$, where $A \in \mathbb{R}^{D \times 2}$ is a random projection matrix. We then use as a summary statistic $A^+ \cdot \bar{\mathbf{x}}_{1:N}$. This allows us to change the dimensionality of the observations $D \in \{2, 5, 10, 50, 100\}$ while keeping both the parameter and summary statistic dimension fixed at the original setting of 2. For the misspecification, we use $\delta = 5, \epsilon = 0.3$. The results for this ablation study is provided in Figure 11. We find that, while the NPE-MDS overall still provides improvement on the robustness compared to both the NPE and NNPE, this gap decreases as the observation dimension increases.

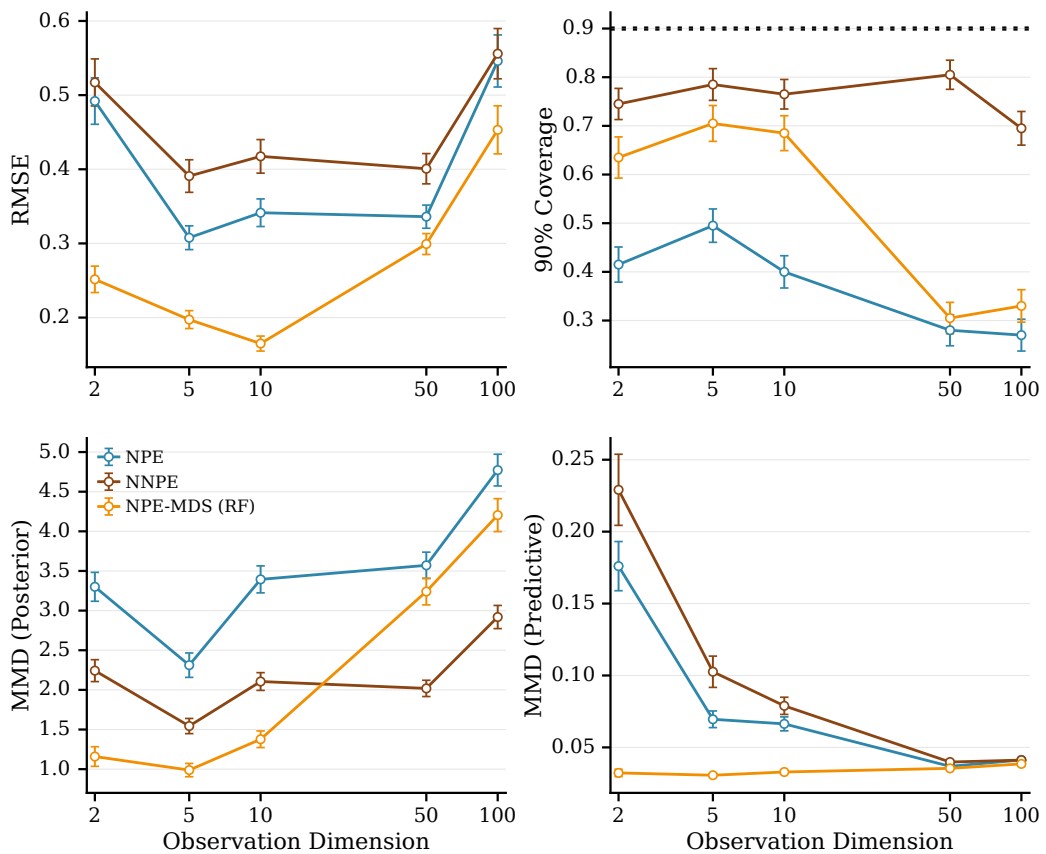

*Figure 11.* Results for ablation with increasing number of dimensions (Appendix D)

### D.3. Optimization Sensitivity Ablation

For the test-time optimization procedure described in section 3.2, we used L-BFGS in our experiments, which was initialized by default at the observed summary $\mathbf{s}_0 = f(\tilde{\mathbf{x}}_{1:N})$. Since equation (2) is a deterministic optimization, we consider its sensitivity to initialization.

With reference to the left plot in Figure 12, we compare the resulting posterior RMSE for four difference initialization schemes. First, using the default observed summary $\mathbf{s}_0 = f(\tilde{\mathbf{x}}_{1:N})$, the observed summary but with Gaussian perturbation, the mean of the simulated training summary statistics, and finally, the zero vector. We observe that, broadly speaking, the mean of the simulated training samples is a better initialization choice measured by the resulting posterior RMSE. This indicates that the optimal minimum-distance summary is likely closer, or more easily reached by the uncontaminated training sample than with the contaminated observed summary, for this particular model and contamination. We further see that as the contamination level increases, this gap widens, which can be interpreted as being due to the additional contamination making the optimization landscape more challenging. We see this as well in the midde plot for Figure 12, which shows the generally increasing standard deviation of the summary distance between the estimated MDS and the oracle summary, using the Gaussian-perturbed default observed summary statistic. While this indicates that our choice of the observed summary statistic as the initialization is not optimal, this result may be dependent on the exact model and contamination setting.

To further investigate how the initialization can be improved, in the right plot of Figure 12, we compare the posterior RMSE using the default single initialization (and optimization procedure) with 10 initializations (and thus 10 resulting optimization procedure). We use the default observed summary statistic, and the 10 initializations are obtained by Gaussian perturbations of this observed summary statistic. We then choose the best resulting MDS from the 10 different optimizations. We observe that while this is a naive approach, it does provides benefits over the default optimization procedure, especially in higher contamination levels. Furthermore, since our optimization is lightweight and runs relatively quickly, the additional computational costs of running 10 optimizations is small (particularly, compared to the training of the decoder mean embedding). Hence, using the multiple restart optimization may be a simple, but powerful approach to improving the optimization procedure.

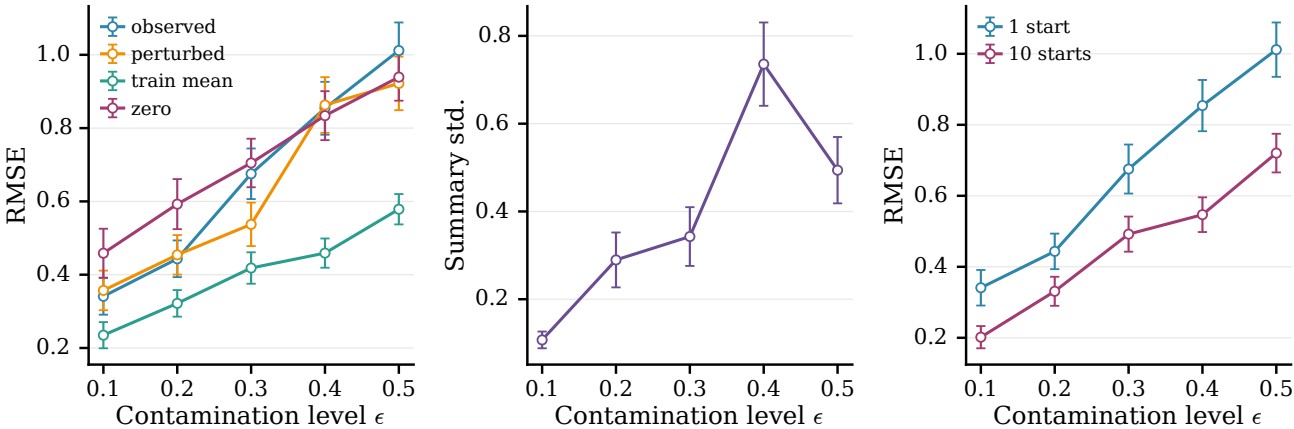

*Figure 12.* Initialization sensitivity of test-time L-BFGS optimization on OUP model Left: posterior RMSE under four initialization settings Middle: standard deviation across Gussian-perturbed initialization, of the summary distance between optimized MDS and oracle summary Right: posterior RMSE comparing MDS optimization with single initialization and multiple restart initialization

### D.4. MMD Kernel Bandwidth Ablation

The MDS objective in Equation (1) depends on the maximum-mean discrepancy, which we use with a Gaussian kernel. It is well-known that the choice of kernel bandwidth is crucial in determining the effectiveness of the Gaussian kernel and the MMD.

To evaluate the sensitivity of the downstream posterior accuracy to the choice of the kernel bandwidth, we scale the default bandwidth choice (which uses the median heuristic) with a multiplicative factor $k$, i.e., $\sigma_k = k \times \sigma_{\mathrm{med}}$. With reference to Figure 13, we find that the median heuristic (corresponding to $k = 1$) is a reliable default choice for our downstream MDS use, although a natural extension would be to select the bandwidth based on a held-out validation set.

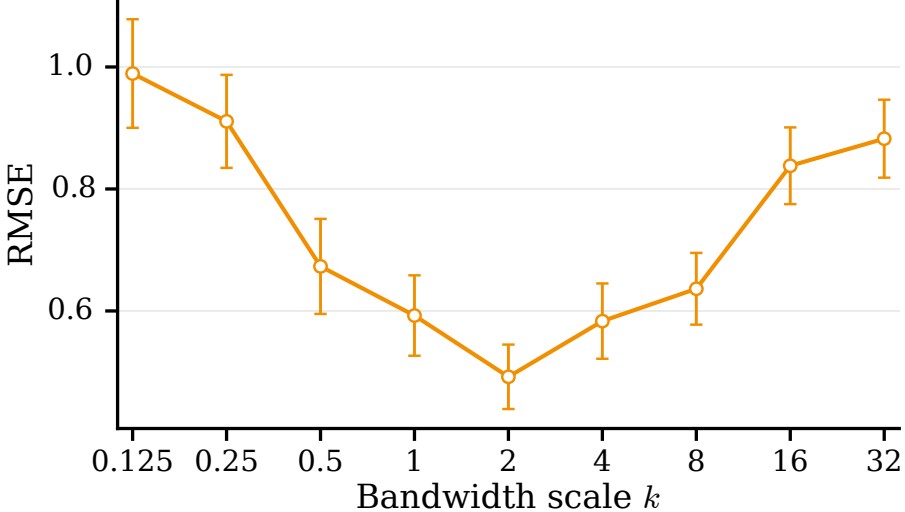

*Figure 13.* Kernel bandwidth sensitivity with the OUP model ($\varepsilon = 0.2$)

## E. NPE with Tabular Foundation Models

NPE-PFN (Vetter et al., 2026) provides an alternate approximation to the Bayesian posterior distribution based on probabilistic foundation models, performing Bayesian inference through in-context learning. In this setting, as we would like to ideally avoid modifying the pretrained NPE-PFN directly, the test-time paradigm is a natural fit, for which the MDS simply adapts the query summary for robustness. We use the same Gaussian task setup as described in Appendix F.1. The results are provided in Figures 14, 15, 16 and 17, and we find that the MDS can provide robustness benefits in this setting.

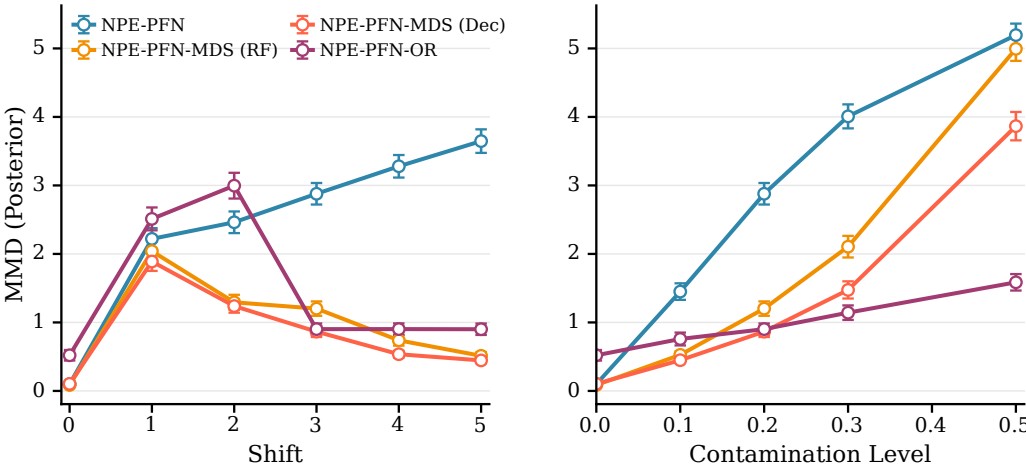

*Figure 14.* MMD between estimated posterior (NPE-PFN) and reference true posterior for Gaussian task (Appendix E)

## F. Further Experimental Details and Results

In this section, we provide further details and additional results, based on the experiments discussed in Section 6.

In all our experiments, we require the training of conditional density estimator for neural posterior estimators and the decoder model proposed in our MDS approach. We use the `sbi` package (Boelts et al., 2025) for the training of the required conditional density estimators. For both the NPE and decoder model, a neural spline flow (Durkan et al., 2019) was used as the density estimator, with default configurations from `sbi`. The Adam (Kingma & Ba, 2014) optimizer was used with a learning rate of $5 \times 10^{-4}$ and the density estimators was trained until convergence.

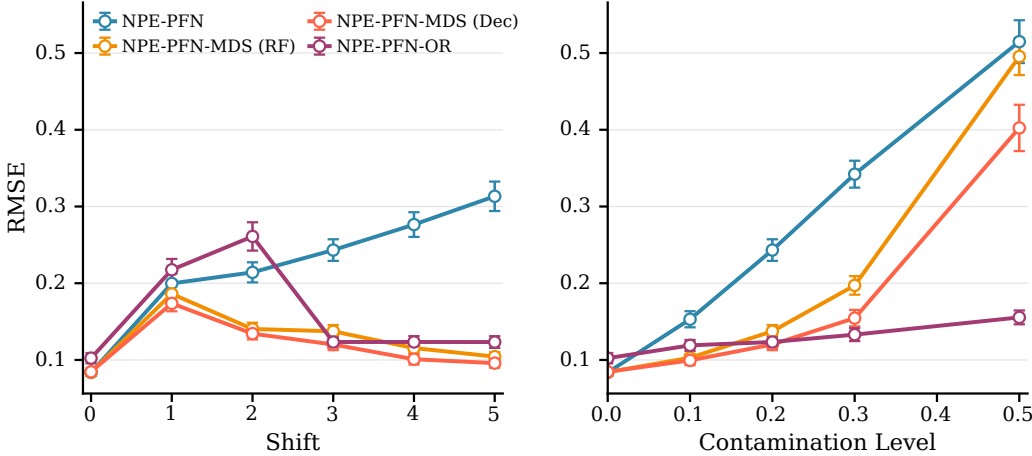

*Figure 15.* RMSE between estimated posterior means (NPE-PFN) and true parameter for Gaussian task (Appendix E)

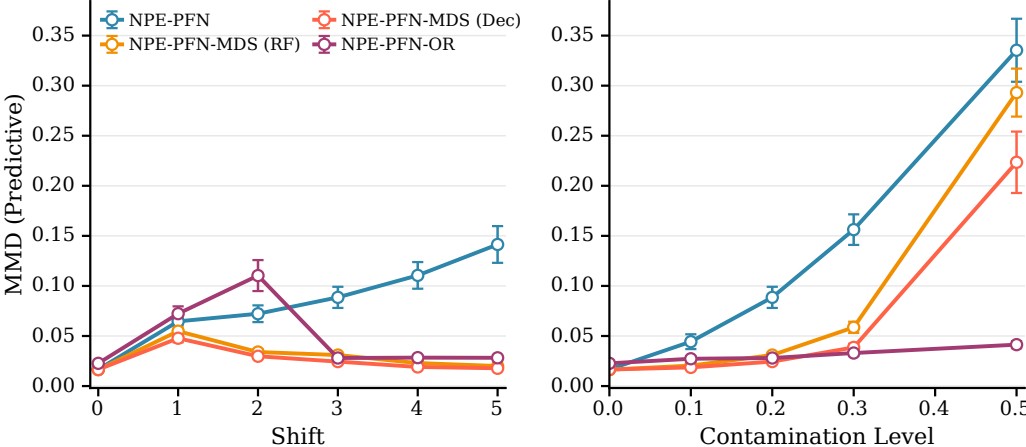

*Figure 16.* MMD between estimated posterior predictive (NPE-PFN) and uncontaminated test dataset for Gaussian task (Appendix E)

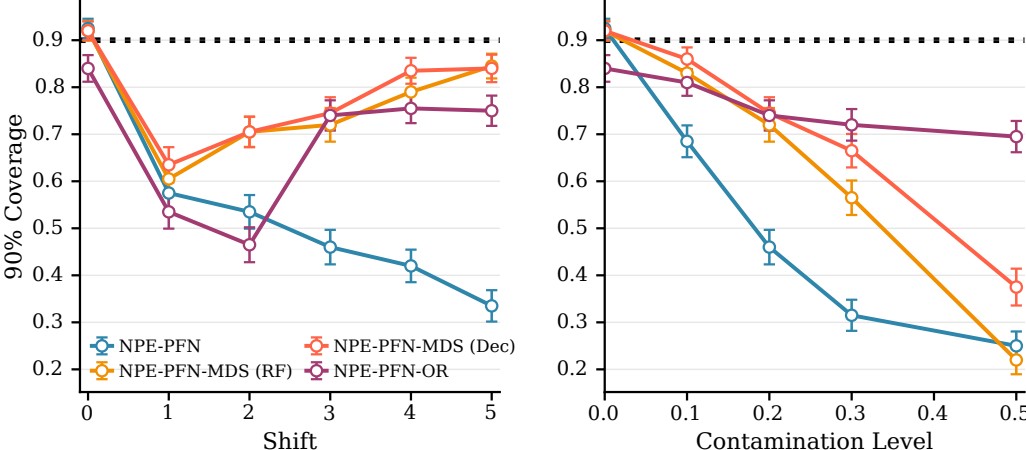

*Figure 17.* Coverage of estimated posterior (NPE-PFN) (Appendix E)

For the MDS algorithm with random Fourier features, the feature approximations corresponding to the RBF kernel were obtained with the `scikit-learn` package (Pedregosa et al., 2011), using the standard median heuristic for the tuning of the bandwidth hyperparameter, and a fixed dimension of 512 for the dimension of the feature space. This feature approximation is used to obtain empirical mean embeddings of training data, which we subsequently use to train a standard fully-connected neural network with 2 hidden layers with 256 units per layer, giving us an amortized conditional mean embedding estimate with respect to $\mathbf{s}$. During test-time, we use the L-BFGS optimization algorithm (Liu & Nocedal, 1989) with line search as implemented in the `PyTorch` package (Paszke et al., 2019) to obtain the adapted MDS in Equation 2. For the decoder model variant of the MDS algorithm, after training of the amortized decoder model with training data, we use the Adam optimizer (Kingma & Ba, 2014) to directly solve an empirical estimate of Equation 1 by drawing samples from the decoder model. As discussed in Appendix C, we reserve 5% of the training data for calibration, which is not used in the pretraining of the conditional mean embedding or decoder model.

For the Noisy NPE of Ward et al. (2022), the spike-and-slab error model was used. Following the notation of $\mathbf{y}$ as the summary statistic and $\mathbf{x}$ as the unobserved latent variable, the error model is (see Equation 8 of Ward et al. (2022)):

$$p(\mathbf{y} \mid \mathbf{x}) = \prod_{j=1}^{D}[(1-\rho)N(x_j, \sigma^2) + \rho\,\text{Cau}(x_j, \tau)]$$

We set $\rho = 1$, $\sigma = 0.01$ and $\tau = 0.2$. As we use the NNPE variant of the algorithm, we directly incorporate this to the simulator model, and train an amortized NPE directly. The NPE-RS method of Huang et al. (2023) was trained by augmenting the standard NPE negative log-likelihood loss function with a regularization term, $\lambda \cdot \text{MMD}^2[f_\phi(\mathbf{x}_{1:N}), f_\phi(\tilde{\mathbf{x}}_{1:N})]$, where $\mathbf{x}_{1:N}, \tilde{\mathbf{x}}_{1:N}$ are training and observed samples respectively. In particular, this requires use of a neural summary statistic $f_\phi$ in order to minimize the regularization term. We use grid search of $\lambda \in \{0.1, 1.0, 5.0, 10.0\}$, and provide the results for the best performing model for each metric. For the OC-SVM (Schölkopf et al., 2001) approach, we fit a one-class SVM using `scikit-learn` (Pedregosa et al., 2011) on training data using the Gaussian kernel with median heuristic, calibrate a threshold using a false positive rate of 5%, mimicking the approach of the MDS, and remove outliers based on the OC-SVM. We then resample from the remaining inliers to maintain the same fixed observations.

In all our tasks, we use datasets $\mathbf{x}_{1:N}$ which $N = 100$, and 100 different test datasets, each corresponding to a true parameter $\theta^*$. The randomness of the error bars in all plots are with respect to the 100 test datasets. For parameter estimation accuracy, we calculate the RMSE over the parameter dimensions, $\sqrt{\frac{1}{d_\theta}\|\bar{\theta} - \theta^*\|_2^2}$, where $\bar{\theta}$ is the posterior mean from the posterior samples for each method. For the calibration of the posterior, we calculate the coverage, by estimating the credible intervals $L_j = Q_{\alpha/2}(\{\theta_j^{(s)}\}_{s=1}^M), U_j = Q_{1-\alpha/2}(\{\theta_j^{(s)}\}_{s=1}^M)$ using posterior samples, and obtain coverage $\mathbf{1}[L_j \leq \theta_j^* \leq U_j]$, which is then averaged over the parameter dimensions. We also obtain posterior predictive metric by comparing the posterior predictive distribution, estimated by resampling from the obtained posterior samples with the simulator model, against the clean, uncontaminated test dataset. Since the MDS approach operates entirely on the summary-space and original data-space, it is natural to compare against the oracle summary, which is the summary statistic of the clean, uncontaminated test dataset, which we do by comparing the Eucliean distance against the obtained MDS. Finally, where possible, such as with the Gaussian task, we compare with the reference posterior directly using the MMD.

### F.1. Gaussian Task

In this section, we discuss the experimental setup for Section 6.1, as well as additional results and experiments based on this setup.

The Gaussian task is based on the following model:

**Prior** $\qquad\qquad\qquad\theta \sim \mathcal{N}_d(\mathbf{0}, I_d)$

**Simulator** $\qquad\qquad\mathbf{x}_{1:N} = (\mathbf{x}_1, \ldots, \mathbf{x}_N),\ \mathbf{x}_i \mid \theta \sim \mathcal{N}_d(\theta, I_d),\ i = 1, \ldots, N$

**Summary statistic** $\quad f_\phi : \mathbb{R}^{N \times d} \to \mathbb{R}^d$

**Dimensionality** $\qquad\ \theta \in \mathbb{R}^d,\ \mathbf{x}_i \in \mathbb{R}^d,\ \mathbf{x}_{1:N} \in \mathbb{R}^{N \times d}$

This provides a closed-form conjugate posterior distribution, which we use to provide an exact measure of the accuracy of our posterior approximation. Note that, for this task, we use a trained neural summary statistic $f_\phi$ (parameterized as a fully-connected neural network), which is learned jointly with the NPE, which allows us to use the NPE-RS approach. We

use 50000 training datasets, and 100 test datasets for evaluation. We use $d = 2$, $N = 100$ unless stated otherwise.

**Misspecification** For a test dataset $\mathbf{x}_{1:N}$ generated from true parameter $\theta^*$, each observation is independently replaced by a shifted outlier with probability $\epsilon$: $\tilde{\mathbf{x}}_i = \mathbf{x}_i$ w.p. $1 - \epsilon$, and $\tilde{\mathbf{x}}_i = z_i \cdot \delta$ w.p. $\epsilon$, where $z_i \sim \mathrm{Unif}\{-1, +1\}$. where $\delta > 0$ controls the outlier magnitude. We vary $\epsilon \in \{0.1, 0.2, 0.3, 0.5\}$ with fixed magnitude $\delta = 3$, and vary $\delta \in \{1, 2, 3, 4, 5\}$ with fixed $\epsilon = 0.2$. This misspecification is based on Experiment 2 of Elsemüller et al. (2025).

In addition to the plot provided in Figure 2, we provide additional plots below. Furthermore, we show the resulting MDS posterior adaptation from four different test datasets in Figure 22.

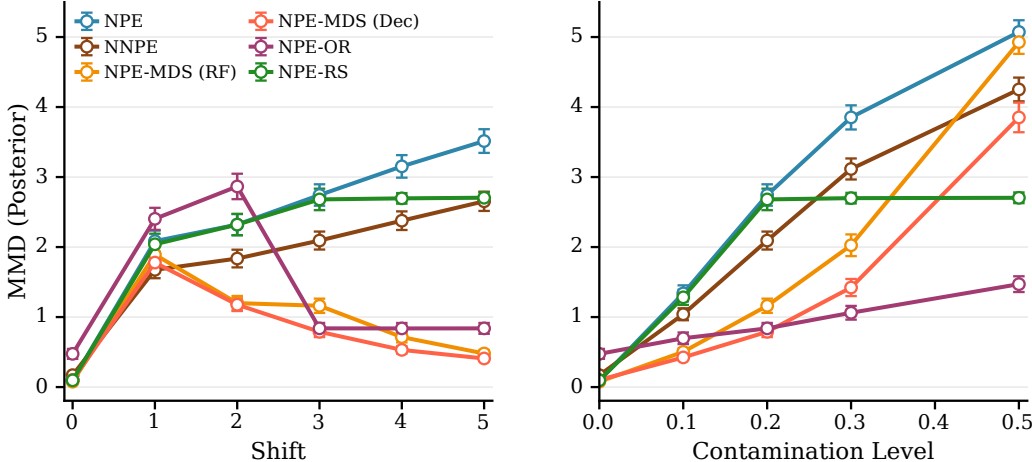

*Figure 18.* MMD between estimated posterior and reference true posterior for Gaussian task (Appendix F.1)

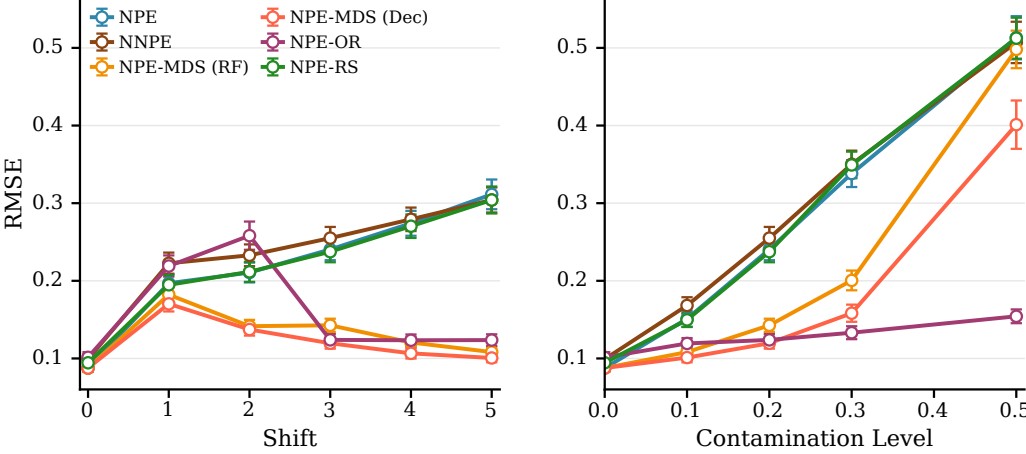

*Figure 19.* RMSE between estimated posterior means and true parameter for Gaussian task (Appendix F.1)

## F.2. OUP Task

The Ornstein–Uhlenbeck process (OUP) is defined by a mean-reverting stochastic differential equation, and is based on the experiment provided in Huang et al. (2023). One key distinction is that we use a fixed summary statistic, instead of a learned neural summary statistic that was done in their setup. The summary statistic $(s_1, s_2, s_3)$ is the mean, variance, and lag-1 autocorrelation respectively, which is averaged over the trajectories and datasets. We use $T = 25$ and $N = 100$ trajectories per dataset. The diffusion variance is set at $\sigma^2 = 0.1$, and initial conditions was set starting from $X_0 = 10$, and the SDE was discretized with a standard Euler-Maruyama scheme. We use 10000 training datasets for the NPE training.

Contamination was induced by replacing a fraction $\epsilon$ of trajectories with contaminated trajectories, generated from a contaminated parameter of $\theta = (-0.5, 1.0)$ and with $\sigma^2 = 0.5$.

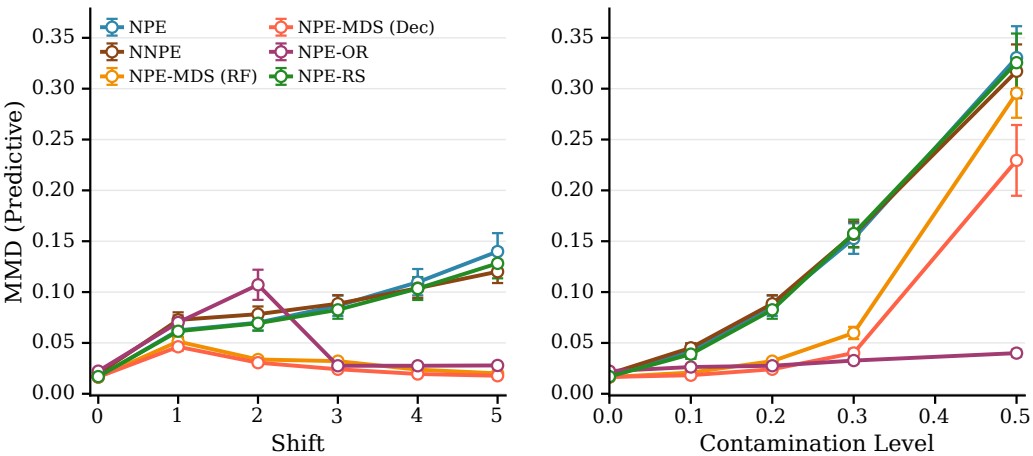

*Figure 20.* MMD between estimated posterior predictive and uncontaminated test dataset for Gaussian task (Appendix F.1)

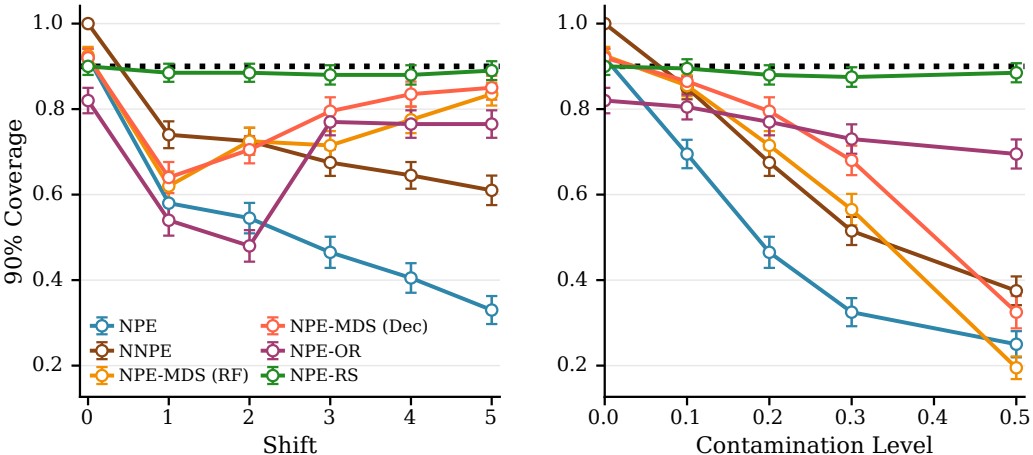

*Figure 21.* Coverage of estimated posterior for Gaussian task (Appendix F.1)

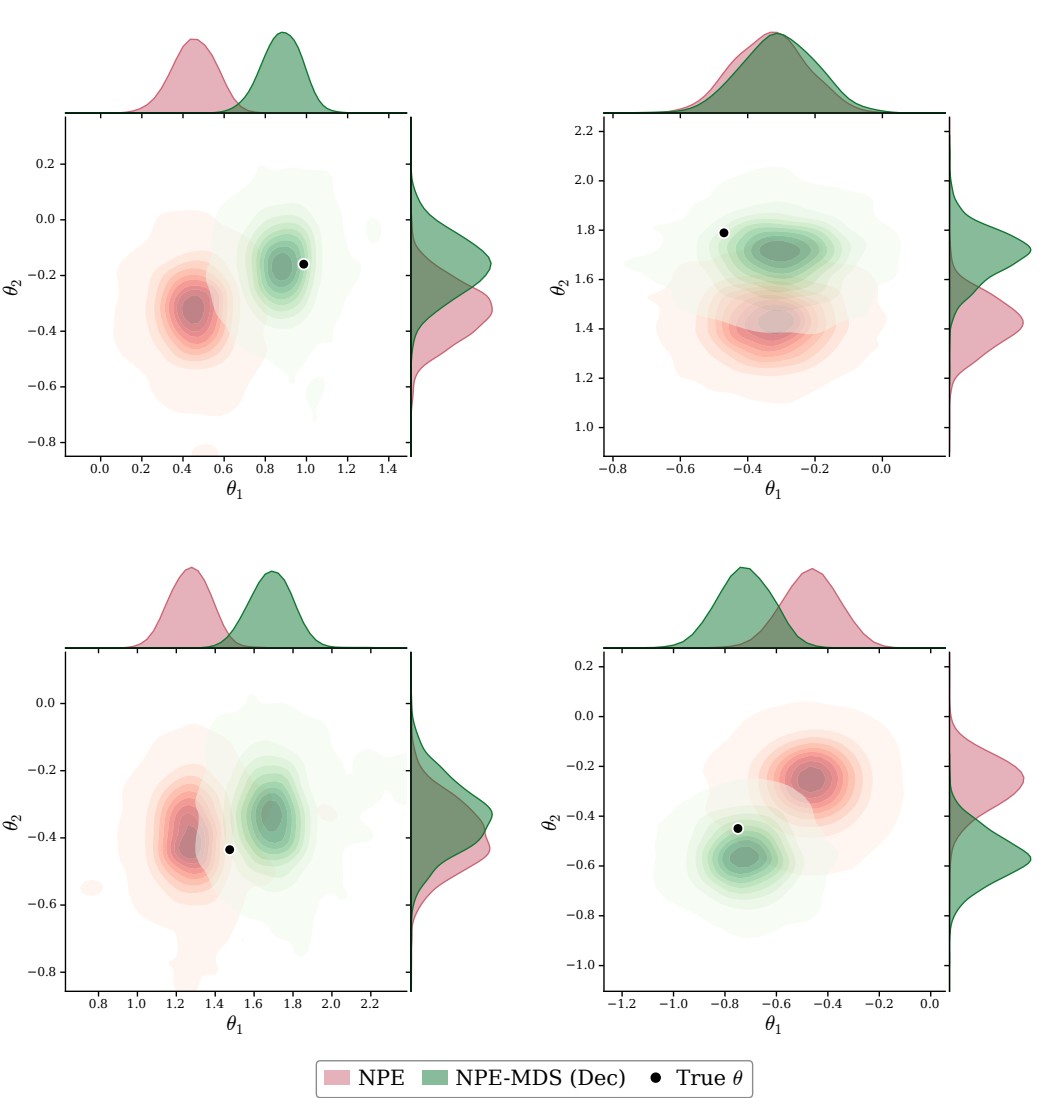

*Figure 22.* Posterior distributions before and after MDS adaptation for Gaussian task, for four test datasets (Appendix F.1)

| | |
|---|---|
| **Prior** | $(\theta_1, \theta_2) \sim \mathrm{U}[0,2] \times \mathrm{U}[-2,2]$ |
| **Simulator** | $dX_t = \theta_1[\exp(\theta_2) - X_t]dt + \sigma\ dW_t,$ |
| **Summary statistic** | $\mathbf{s}(\mathbf{x}_{1:N}) = (s_1, s_2, s_3)$, where $\mathbf{x}_{1:N} = \mathbf{X} \in \mathbb{R}^{N \times T}$: |

$$s_1 = \frac{1}{NT}\sum_{n=1}^{N}\sum_{t=1}^{T}\mathbf{X}_{n,t}$$

$$s_2 = \frac{1}{NT}\sum_{n=1}^{N}\sum_{t=1}^{T}(\mathbf{X}_{n,t} - s_1)^2$$

$$s_3 = \mathrm{corr}\Big(\mathrm{vec}(\mathbf{X}_{:,1:T-1}),\ \mathrm{vec}(\mathbf{X}_{:,2:T})\Big).$$

| | |
|---|---|
| **Dimensionality** | $\theta \in \mathbb{R}^2$, $\mathbf{x}_i \in \mathbb{R}^T$, $\mathbf{x}_{1:N} \in \mathbb{R}^{N \times T}$ |

We provide the results and posteriors for the OUP task in Figures 23 and 24.

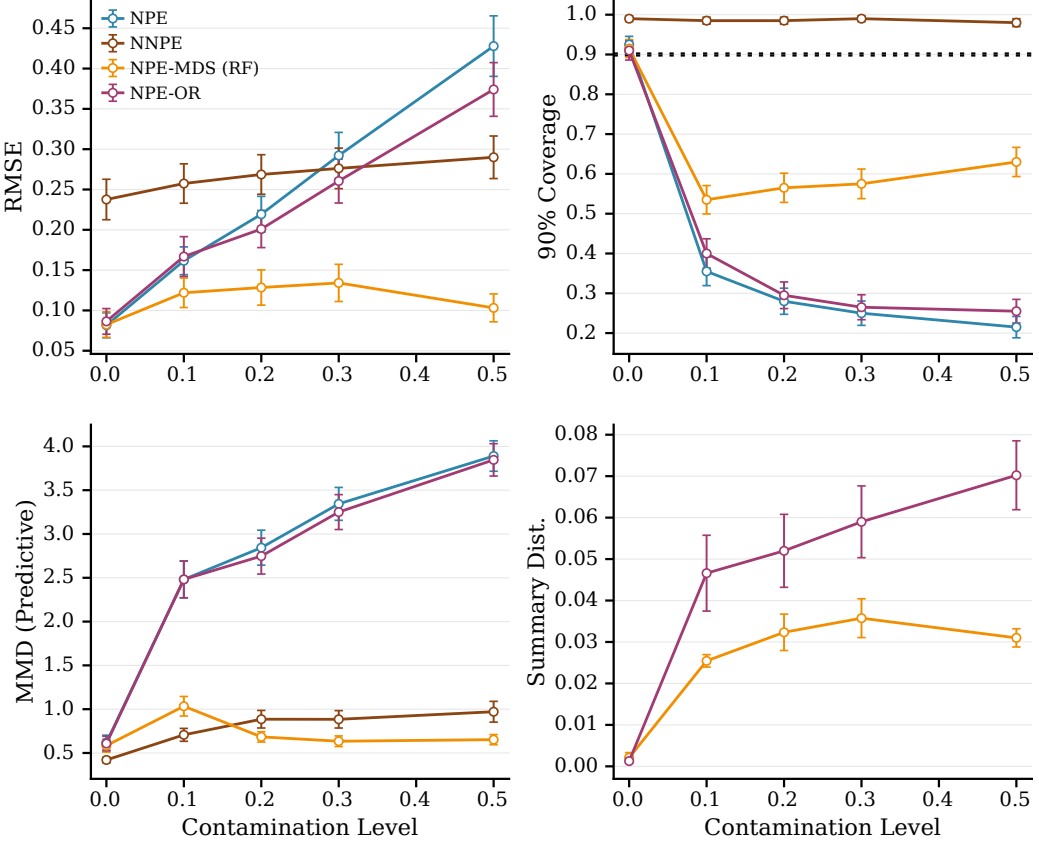

*Figure 23.* Results for OUP task (Appendix F.2)

### F.2.1. OUP WITH LEARNED NEURAL SUMMARY STATISTIC

In this section, we provide results for the OUP model, but with a learned encoder instead of a fixed summary statistic, which was used previously. Specifically, following the learned summary statistic procedure done in Appendix F.1, we first train an encoder $f_\phi(\mathbf{x})$ and the neural posterior estimator jointly in a standard NPE pipeline, before freezing both the encoder and the NPE for downstream use with the MDS, essentially replacing the fixed summary statistic with the frozen encoder.

With reference to Figure 25, we see that the MDS is able to preserve the robustness benefits over the standard NPE baseline in this learned summary statistic setting, similar to the fixed summary statistic setting discussed previously.

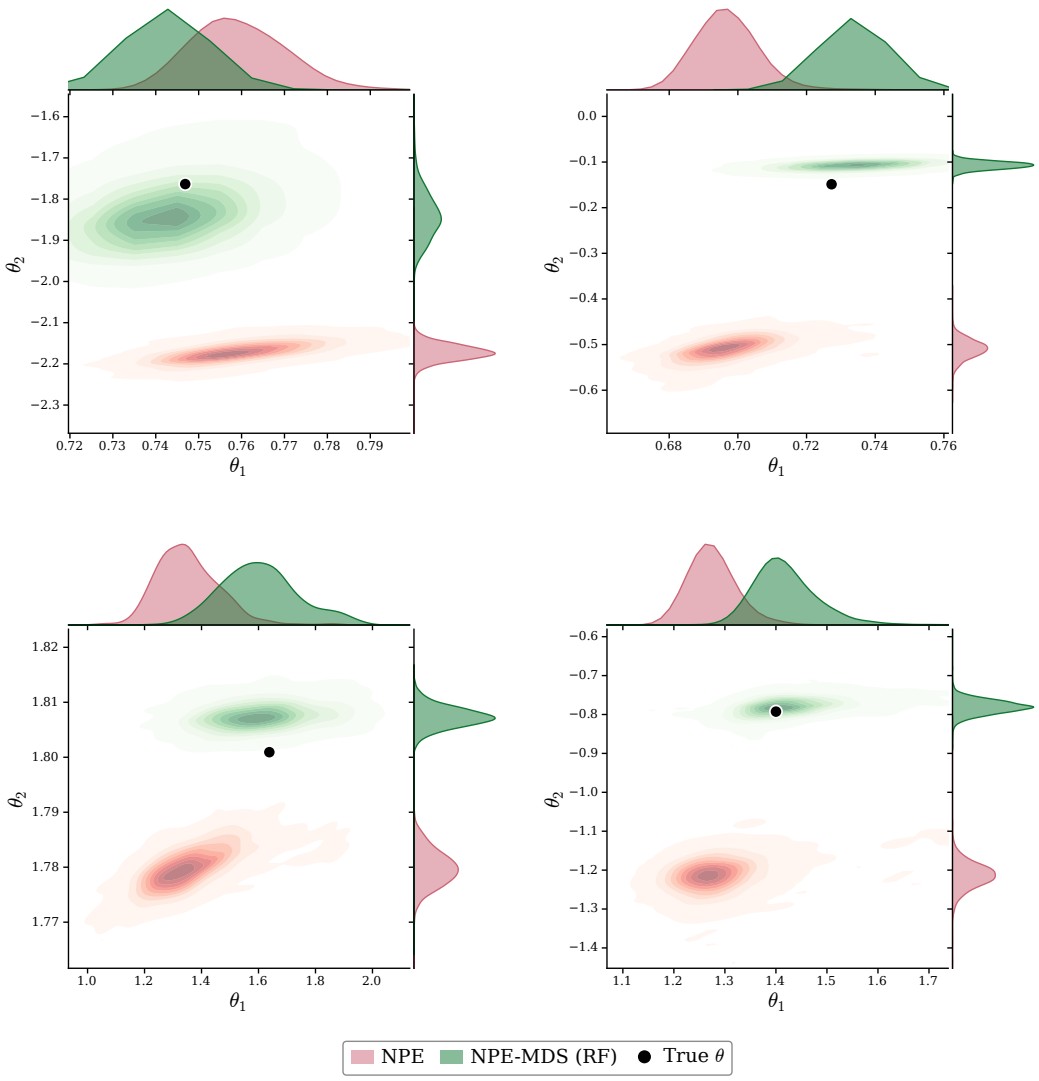

*Figure 24.* Posterior distributions before and after MDS adaptation for OUP task for four test dataset (Appendix F.2)

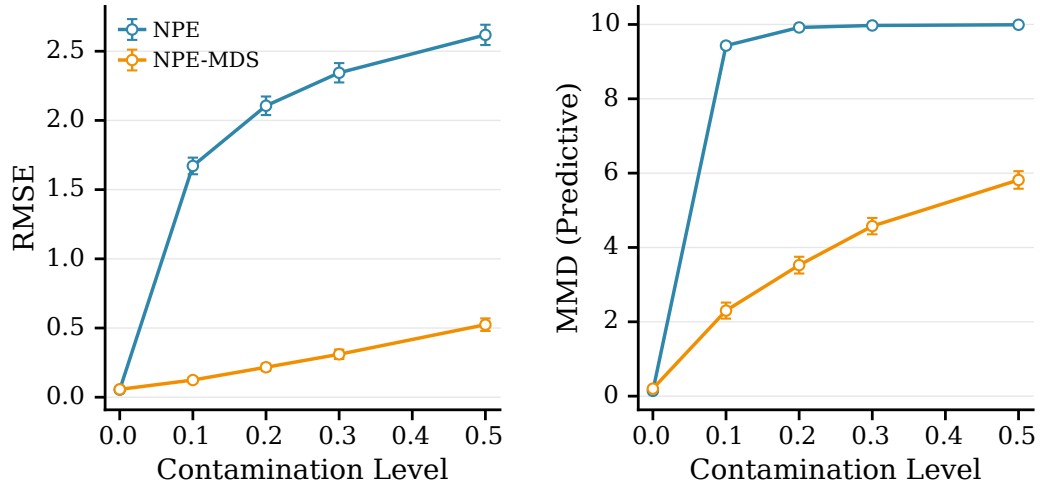

*Figure 25.* OUP model with learned summary statistic

## F.3. SIR Task

The SIR (Susceptible-Infected-Recovered) model is a classic epidemiological model which models the spread of infectious disease through a population. This setup is similar to Ward et al. (2022). This particular SIR variant is a stochastic variant, which has standard ODE dynamics for $S, I, R$ and a mean-reverting diffusion process for the reproduction number $R_0$, which is coupled to the ODE through the effective transmission rate $\beta_{\text{eff}} = \gamma \cdot R_0$. In the diffusion process, for $R_0$, we have $\bar{R}_0 = \beta/\gamma$ and set $\sigma = \eta = 0.05$ The system is initialized with $(S_0, I_0, R_0) = (0.999, 0.001, 0)$, and integrated with a standard Euler-Maruyama scheme over $T = 365$ days, giving daily infection counts $X_t = N \cdot I_t$ where the population $N = 10000$. We simulation 100 trajectories for each parameter, and the parameters to be inferred are the infection and recovery rate, $\beta, \gamma$ respectively. We use 10000 training datasets for the NPE training.

Following Ward et al. (2022), for the summary statistic $(S_1, \ldots, S_6)$, we take this to be the log mean infections, log median infections, log peak infections, log day of peak, log day when 50% of cumulative infections are reached, and the lag-1 autocorrelation, which is averaged across trajectories.

Model misspecification is introduced through weekend underreporting, for a fraction $\epsilon$ of the trajectories, weekend infection counts are undereported (by 5%), and redistributed to the following Monday.

**Prior**
$$(\beta, \gamma) \sim \text{U}\big[\{(\beta, \gamma) : 0 < \gamma < \beta < 0.5\}\big].$$
$$\frac{dS}{dt} = -\beta_{\text{eff}}(t)SI,$$

**Simulator**
$$\frac{dI}{dt} = \beta_{\text{eff}}(t)SI - \gamma I,$$
$$\frac{dR}{dt} = \gamma I,$$
$$dR_0 = \eta(\bar{R}_0 - R_0)dt + \sigma\sqrt{|R_0|}\,dW_t$$

**Summary statistic**  $\mathbf{S}(\mathbf{x}_{1:N}) = (S_1, \ldots, S_6)$, where $\mathbf{x}_{1:N} = \mathbf{X} \in \mathbb{R}^{N \times T}$

**Dimensionality**  $\theta \in \mathbb{R}^2$, $\mathbf{x}_i \in \mathbb{R}^T$, $\mathbf{x}_{1:N} \in \mathbb{R}^{N \times T}$, $\mathbf{S} \in \mathbb{R}^6$.

We provide results and posteriors for the SIR task in Figures 26 and 27.

### F.3.1. SIR UNDER FURTHER MISSPECIFICATION

While the theoretical robustness guarantees provided in Section 4 is for the Huber contamination model and is typically interpreted with modest values of contamination ($\varepsilon < 0.5$), given that the MDS objective is not tied to outlier contamination specifically and instead adapts the summary statistic by matching the data distribution to the observed empirical distribution, this suggests that MDS may be able to provide robustness benefits beyond the Huber contamination model setting.

To evaluate this behavior, we consider further extending the SIR benchmark to larger contamination settings. Since the misspecification in this benchmark is structural, we can consider increasingly large fractions of contamination, where at $\varepsilon = 1.0$, all observed trajectories are distorted by the weekend-reporting delay.

As we can see from Figure 28, MDS improves over the standard NPE baseline and NPE-OR, and the improvement continues to improve as the contamination fraction reaches 1.0. Thus, this provides empirical evidence that MDS can retain robustness benefits even under structured misspecification.

## F.4. Cryo-EM Task

Cryo-electron microscopy (cryo-EM) is an imaging technique that captures projected 2D images of 3D biomolecules, whose configuration (conformational state) varies across images. In particular, this problem setting is especially challenging due to the presence of nuisance variables, such as the orientation of the image and measurement noise. We refer the interested reader to Dingeldein et al. (2025); Evans et al. (2025) for further details about this problem setting. We use the HSP90 (Heatshock protein 90) model, provided as a benchmark in the `cryoSBI` package in (Dingeldein et al., 2025). For each conformational state $\theta$, we simulate a $32 \times 32$ image, and the goal is to estimate a conformational index $\theta$ with prior as discrete uniform over $K = 20$ indices. As summary statistic, we compute the per-image skewness, kurtosis and intensity range, and aggregate this across the dataset of $N = 100$ images, and use the resulting mean and standard deviation, giving a 6 dimensional summary statistic. We use 15000 training datasets for the NPE training.

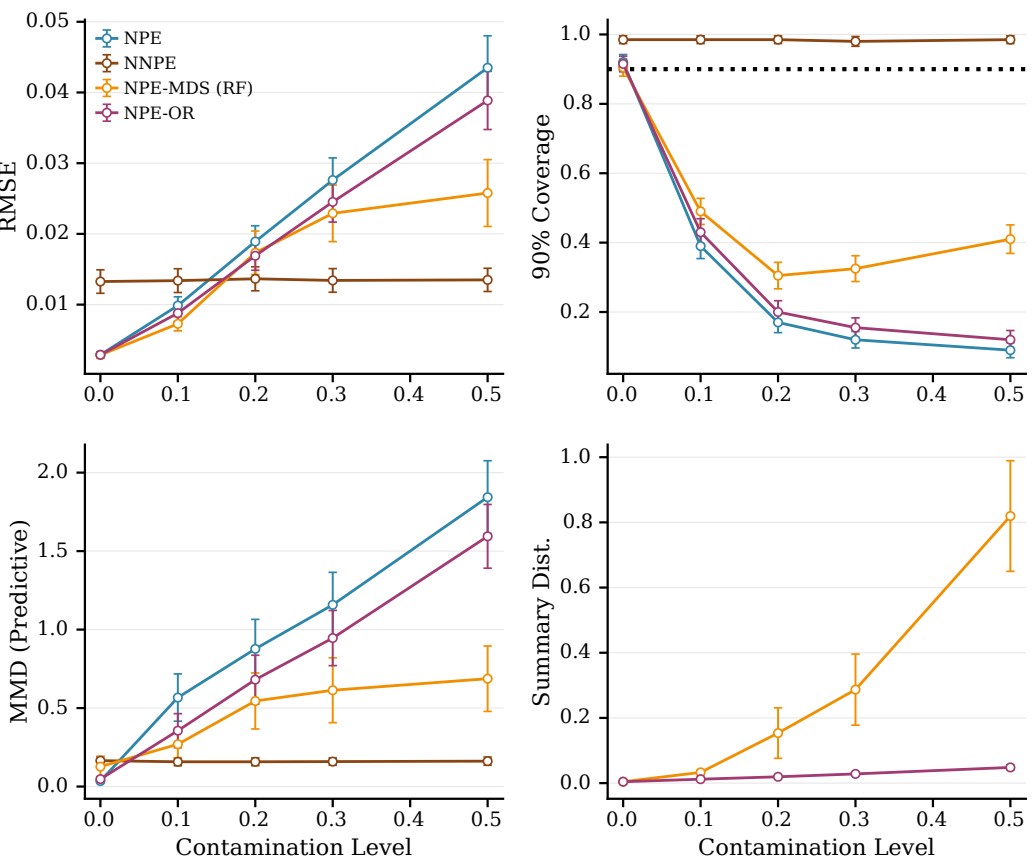

*Figure 26.* Results for SIR task (Appendix F.3)

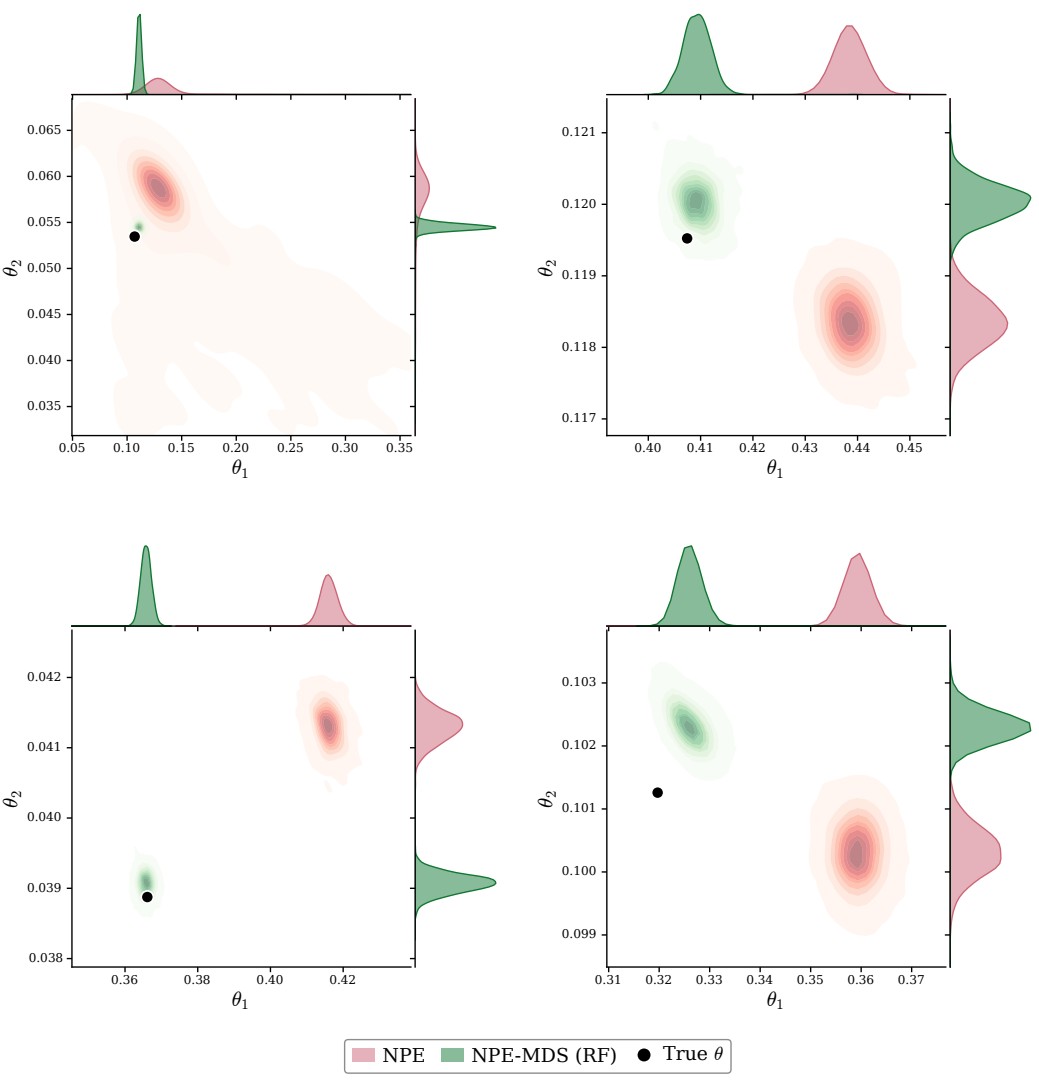

Figure 27. Posterior distributions before and after MDS adaptation for SIR task for four test dataset (Appendix F.3)

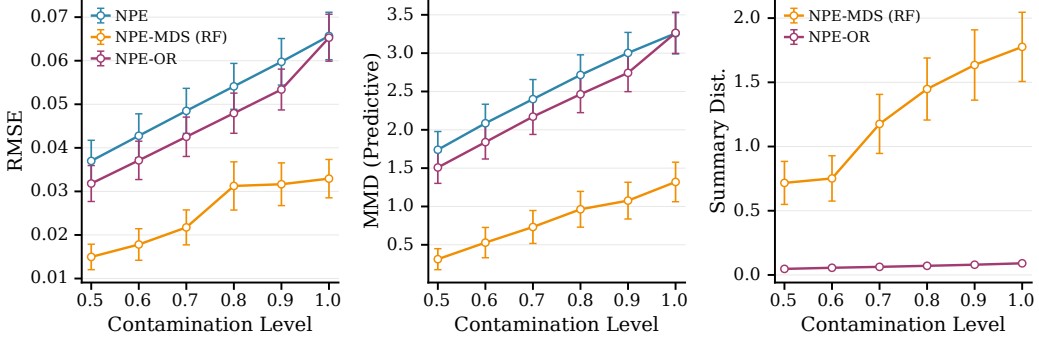

Figure 28. Results for SIR model under increasing misspecification

Model misspecification is induced by contaminating $\epsilon$ proportion of the images in the dataset with pure Gaussian noise, which simulates measurement or equipment error that is common in realistic cryo-EM settings.

We provide results and posteriors for the cryo-EM task in Figures 26 and 27.

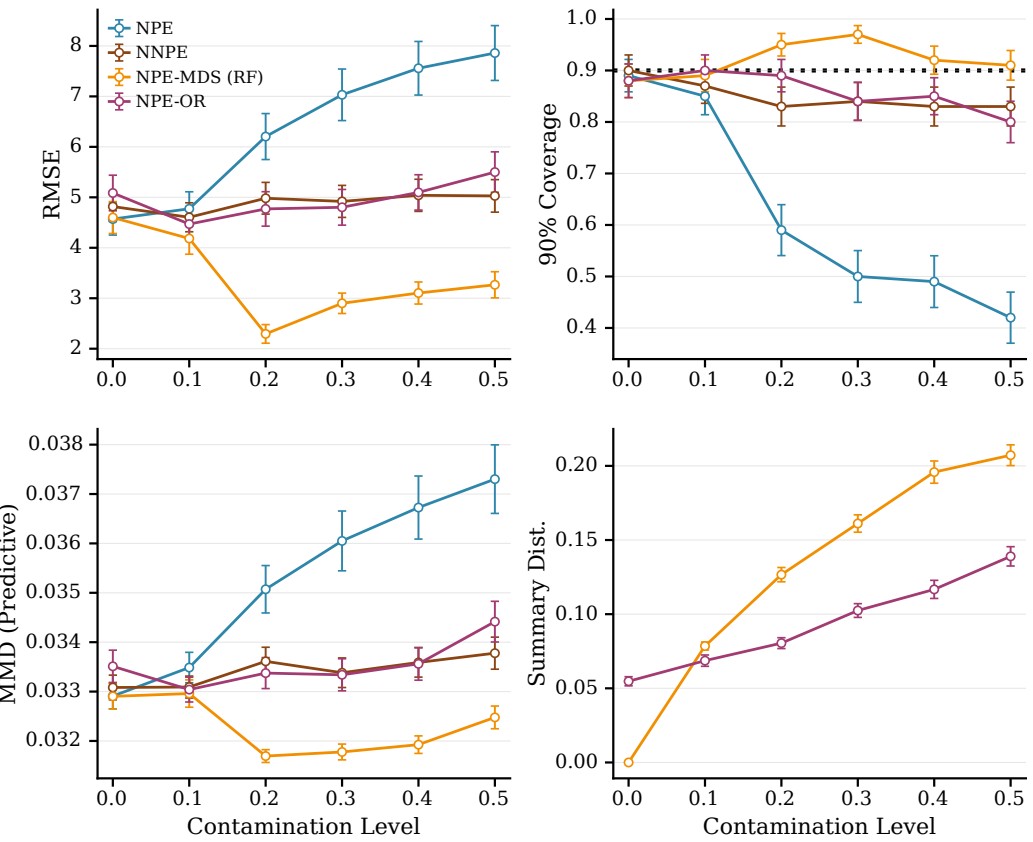

*Figure 29.* Results for cryo-EM task (Appendix F.4)

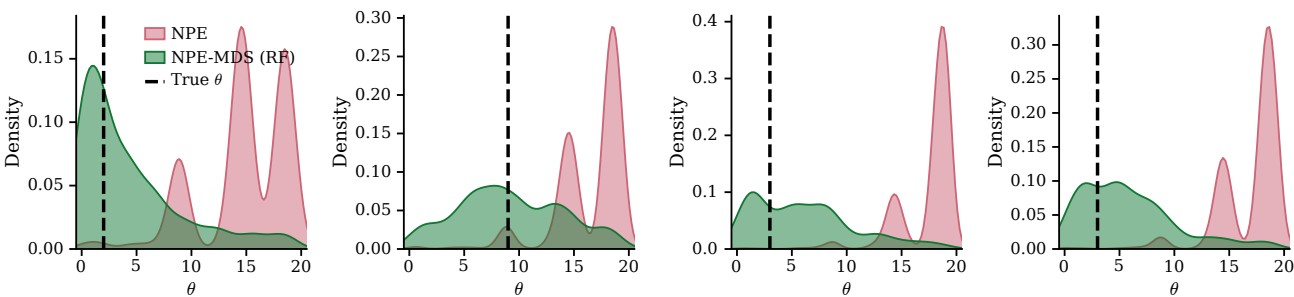

*Figure 30.* Posterior distributions before and after MDS adaptation for cryo-EM task for four test dataset (Appendix F.4)

