# OpenReview forum: "Minimum Distance Summaries for Robust Neural Posterior Estimation"
_ICML.cc/2026/Conference — ICML 2026 regular_

### Official Review · Reviewer_dXdi · 2026-03-09

**Soundness:** 3
**Presentation:** 3
**Significance:** 3
**Originality:** 3
**Overall Recommendation:** 5
**Confidence:** 3

**Summary:**

This paper proposes a method SBI is done using two-steps; first get a summary, and then predict the posterior. To combat model misspecification, at test time, the summary is optimized by minimizing the MMD in the data-space, to make the SBI posterior robust against contamination.

**Compliance With Llm Reviewing Policy:**

Affirmed.

**Final Justification:**

I am happy with the responses of the authors. I also checked the other reviews. I am much in favor of acceptance as I see no major issues with this work.

**Key Questions For Authors:**

0. Is it possible to perform also more experiments with learned summary functions? Only the bivariate case has learned summaries - why is that? I would be interested to know how the performance is for learned summaries in some of the other cases.

1. Regarding the background on SBI; is it true that SBI is _only_ used in case when p(x) is intractable to evaluate? Because my understanding is that SBI (e.g. PFN or NP) is also used when the posterior itself is intractable (and maybe p(x) can be tractable to evaluate).

2. I don't understand the remark in line 212 - 214 about x_{1:N}. Since we only observe the dataset x tilde, what is "the dataset x"? Is that the distribution p(x)?

3. "Since our goal is to obtain a robust SBI approach, we require a robust divergence, which makes the KL unsuitable." line 209 (second column). What result are you referring to here? Can this be backed up with a reference? Or provide some insight here; that would be very welcome.

4. Figure 6; how can the performance improve for more contamination?

5. "but requires a computationally expensive Gram matrix inversion, which is especially problematic if the number of training samples
are large." I fail to see that the inversion is necessary. You could use for example the estimators in
Gretton, A., Borgwardt, K. M., Rasch, M. J., Schölkopf, B., & Smola, A. (2012). A kernel two-sample test. The journal of machine learning research, 13(1), 723-773.
Without the need to perform any inversion?

6. I don't understand how the calibration of the misspeification detector is done; doesn't this require contaminated samples? Yet I thought before you assumed you did not have access to such samples during traintime?

7. For me, it's not clear what is the y-axis of Figure 2. Also for the figures in the appendices, the y-axis is unclear. MMD Predictive, MMD Posterior, etc. were not defined ?

8. In lines 346-351, two baselines are mentioned, NNPE and NPE-OR; but it seems these both refer to the method of Ward (?). What is their difference? It would be good to have consistent names for the related works, already introduced in the related work section. I feel the paper would benefit a lot if these competing approaches are described in more detail.

**Limitations:**

There is a good discussion noting some of the limitations

**Strengths And Weaknesses:**

I could not check the correctness of the proofs.

The presentation is generally good, but the references seem off or ill-formed, often missing years and having wrong authors.

The paper is highly significant with the rapidly growing popularity of SBI and PFN based methods. I am very interested in developing applications using the results of this paper.

The contribution is original as far as I could tell.

The idea is novel to my knowledge and strong.

---

> ### Author Rebuttal · Authors · 2026-03-31
>
> Dear Reviewer dXdi:
>
> We thank the reviewer for their positive feedback, and are encouraged that they found our work to be novel and applicable.
>
> 0. Yes, MDS is fully compatible with learned summaries. Once the encoder has been trained, we can treat the output exactly like a fixed summary statistic. In Table 1, we provide additional experiments of the OUP model with a learned neural encoder, and observe that the robustness benefits of MDS are preserved. We will include this experiment, and corresponding details in the revision.
>
> | $\varepsilon$ | NPE | NPE-MDS |
> |---|---:|---:|
> | 0.0 | **0.640** (0.039) | 0.642 (0.039) |
> | 0.1 | 1.072 (0.046) | **0.642** (0.039) |
> | 0.3 | 1.159 (0.034) | **0.661** (0.039) |
> | 0.5 | 1.429 (0.043) | **0.801** (0.040) |
>
> Table 1: RMSE against the true parameter in the OUP model with learned neural summary statistic (standard error in parentheses)
>
> 1. We agree that our wording was too narrow. While SBI is most commonly motivated with intractable likelihoods, amortized posterior estimators are broadly useful beyond this setting, particularly when repeated inference across many datasets may be desirable, even when the likelihood is available. We will revise the manuscript to clarify this distinction.
>
> 2. $\mathbf{x}\_{1:N}$ is intended to denote a generic model-generated dataset, as opposed to the observed dataset $\tilde{\mathbf{x}}\_{1:N}$. Note that in Equation 1, the objective uses $\mathbb{P}\_{\mathbf{x} \mid \mathbf{s}}$, the conditional distribution for a single data point (e.g. $\mathbf{x}\_1$), as opposed to over the entire dataset $\mathbf{x}\_{1:N}$. This is necessary as during test-time, we only have a single realization of the test dataset $\tilde{\mathbf{x}}\_{1:N}$. We will rewrite this paragraph with improved clarity.
>
> 3. The forward-KL divergence corresponds to the maximum likelihood objective, and is not considered robust to contamination as the log-likelihood is unbounded, and hence a small fraction of outliers with small model density can dominate the objective. A closed-form example showing the robustness of the MMD compared to the KL is given in Appendix D of Chérief-Abdellatif and Alquier (2020). We will revise the manuscript to make this point more explicit.
>
>
> 4. We note that Figure 6 varies the contamination shift $\delta$ at a fixed contamination fraction $\varepsilon$. If we increase $\varepsilon$ instead, we find that the performance degrades as expected.
>
> Specifically, the behavior in Figure 6 is due to the bounded RBF kernel used in the MMD for MDS. As the outlier points move further away from the inliers, the influence of outliers is diminished, and so large-shift outliers can affect the MDS less than moderately shifted outliers.
>
> Since the sensitivity to outliers is controlled by the kernel bandwidth, we ran an additional bandwidth ablation (Table 2). By scaling the default bandwidth (median heuristic) with a scaling factor $k$, we find that the improvement in RMSE with increasing $\delta$ is strongest for small scaling factors, and weaker for larger ones. We will clarify this point more clearly in the revised manuscript.
>
> | $\delta$ | $k=0.25$ | $k=4$ | $k=16$ |
> |:------------------------------|:----------------|:----------------|:----------------|
> | 0.1 | 0.273 (0.053) | 0.209 (0.040) | 0.188 (0.035) |
> | 1.0 | 0.157 (0.035) | 0.165 (0.039) | 0.175 (0.037) |
> | 5.0 | 0.092 (0.026) | 0.086 (0.019) | 0.132 (0.031) |
>
> Table 2: RMSE against true parameter in Gaussian model with increasing contamination shift $\delta$ across different scaling factor $k$ (standard error in parentheses)
>
>
> 5. While the MMD estimators in Gretton et al. (2012) do not require matrix inversion, our point concerned estimating the summary-conditional mean embedding. Kernel conditional mean embedding typically involves inverting a Gram matrix, e.g., Theorem 5 of Song et al. (2009) and Equation 10 of Song et al. (2013).
>
> 6. Calibration does not require the contaminated test dataset as it is performed using the correctly specified training dataset. We use only held-out, simulator generated samples to compute the null distribution of the test statistic, and set $\tau$ as its $1-\alpha$ quantile. At test-time, adaptation is only done if the test statistic on the observed dataset exceeds $\tau$. This is discussed in section C of the appendix, and we will clarify this procedure more explicitly in future revisions.
>
> 7. We will revise the manuscript so that figure labels and definitions are more clearly defined. Posterior MMD and predictive MMD refers to the MMD between the estimated posterior and the reference posterior, and MMD between the estimated posterior-predictive and uncontaminated test dataset respectively.
>
> 8. We will improve on the clarity of the different baseline methods in the revision. Only NNPE refers to the method of Ward et al. (2022), while NPE-OR refers to a baseline which uses outlier removal to preprocess test datasets, elaborated in section D of the appendix.

---

> > ### Author Rebuttal · Reviewer_dXdi · 2026-04-03
> >
> > I am happy with the responses of the authors. I also checked the other reviews. I am much in favor of acceptance as I see no major issues with this work.

---

### Official Review · Reviewer_PqPG · 2026-03-12

**Soundness:** 3
**Presentation:** 3
**Significance:** 3
**Originality:** 3
**Overall Recommendation:** 5
**Confidence:** 4

**Summary:**

The paper proposes an approach to address model misspecification when conducting simulation-based inference (SBI) via a neural posterior estimator (NPE). The misspecification may arise from observed data deviating from the simulator's prior predictive distribution, e.g., due to measurement noise, outliers, operator/equipment calibration/error, etc. Existing approaches address this by coupling robustness into the SBI engine, e.g., via modifications to the NPE training objective, or leveraging observed data during training. As the authors correctly point out, this hinders the primary benefit of amortisation - the ability to reuse the single pre-trained network.
To address this, the proposed minimum distance summaries (MDS) approach adapts the queried summary statistic *post-hoc*, to minimise the maximum mean discrepancy (MMD) between the observed data and a summary conditional predictive distribution (learned via a decoder network). To enable computational efficiency, random Fourier features are used to approximate the Gaussian kernel feature map. This yields a simple MSE regression to predict the finite-dimensional kernel mean embedding. Consequently, no re-training of the NPE model is needed, with MDS providing a lightweight model-free post-hoc robustness enhancement.

**Compliance With Llm Reviewing Policy:**

Affirmed.

**Final Justification:**

Based on the initial submission, the author responses during the rebuttal and other reviewers' evaluation, I believe there is consensus on acceptance for this paper. The authors addressed my questions well and reinforced my initial evaluation.

**Key Questions For Authors:**

1. Could MDS be adapted or extended to account for severe structural model misspecification (as opposed to data corruption)? Perhaps the Huber model is permissive enough of even such kinds of corruption (up to a limit)?
2. How sensitive is the test-time L-BFGS optimization to initialisation? Did the authors encounter any difficulties with optimisation leading to local minima in their experiments?
3. How sensitive is the downstream posterior accuracy to the choice of the random Fourier features' kernel bandwidth, and are there best practices for auto-tuning this for a black-box simulator?

**Limitations:**

Yes

**Strengths And Weaknesses:**

**Soundness**:
- Strengths
	- The approach is principled and fundamentally sound, with formal guarantees provided.
	- Decoupling robustness from NPE is a strong conceptual contribution, allowing practitioners to leverage massive pre-trained inference frameworks, and enhance robustness *post-hoc* for potential distribution shifts in observed data.
	- Leveraging random Fourier features to avoid full conditional density estimation is elegant, pragmatic, and effective. This way MDS bypasses the computational expense typically associated with estimating MMDs and flow-based decoders.
	- Since MDS learns a summary-conditional mean embedding, the MMD objective can also be used as a cheap two-sample test for misspecification.
- Weaknesses
	- For complex, high-dimensional summary spaces, the optimisation landscape (encountered when adapting summary $s^*$ using L-BFGS to minimise MMD) will likely be non-convex. A discussion on risks of being trapped in local minima (or not) could be good to incorporate.
	- This is not really a weakness per-se, but an observation that the adaptation performance hinges critically on the accuracy of the decoder network (that maps $s$ to MMD embedding $\mu$). So it feels one must be careful in training the decoder network.

**Significance**: The paper is of high significance and relevance to modern SBI researchers and practitioners as real-world data is inherently noisy. MDS is a principled approach to enable robustness that brings flexibility via its *post-hoc* nature (no need to re-train the NPE model) and computational efficiency (it is lightweight). It can therefore have  significant impact on posterior estimation accuracy for noisy datasets (e.g., biological data).

**Presentation**: The paper is well written and presented overall. There are only minor issues which I list below.

**Originality**: The proposed MDS approach is novel in its concept and design. Decoupling inference engines from robustness adaptation is a pragmatic and effective design.

---

*Minor issues*: some of the papers on arxiv may have peer-reviewed versions available. I recommend double-checking the references list once. Müller et. al. is missing the publication year. Some of the links are going over the column boundaries (but perhaps this is a template-specific issue).

---

> ### Author Rebuttal · Authors · 2026-03-31
>
> Dear Reviewer PqPG:
>
> We thank the reviewer for their positive feedback, and are encouraged that they found our work to be novel and highly significant.
>
> 1. We focused primarily on the Huber model of contamination, and our theoretical guarantees are formulated under this setting. However, while our guarantee is local in the contamination fraction $\varepsilon$, empirically, our MDS procedure does well for a range of $\varepsilon$ values.
>
> In particular, in the SIR experiment, the type of misspecification is a form of **structured misspecification** where weekend counts are systematically underreported. We extend the experiment with increasing misspecification $\varepsilon$, where at $\varepsilon = 1$, all observed trajectories are distorted by the structured misspecification. Even in this regime, we find that MDS continues to provide robustness benefits, improving over the standard NPE baseline across all fractions of contamination (Table 1), providing empirical evidence that MDS is able to improve robustness even under **systematic, structured misspecification**.
>
> We will revise the paper to make the type of misspecification more explicit, as well as include further experiments with structured misspecification for further evaluation.
>
> | $\varepsilon$ | NPE | NPE-MDS |
> |---|---:|---:|
> | 0.6 | 0.0502 (0.0049) | **0.0271** (0.0042) |
> | 0.7 | 0.0563 (0.0051) | **0.0356** (0.0051) |
> | 0.8 | 0.0621 (0.0052) | **0.0455** (0.0057) |
> | 0.9 | 0.0677 (0.0053) | **0.0512** (0.0064) |
> | 1.0 | 0.0730 (0.0053) | **0.0593** (0.0082) |
>
> Table 1: RMSE against true parameter in SIR model with increasing misspecification (standard error in parentheses)
>
>
> 2. We agree that the objective is non-convex so global optimality is not guaranteed. Empirically, we find that across all experiments, the L-BFGS optimization converges quickly (10 to 30 iterations), and we did not observe any instability issues.
>
> To assess the sensitivity of the optimization to initialization, we ran an additional ablation study on the OUP model, comparing three different initializations: the observed summary $\mathbf{s}_0 = \tilde{s}$ used in our experiments, Gaussian perturbations to $\mathbf{s}_0$, and the zero vector.  We find that, for mild contamination levels, the downstream posterior accuracy remains relatively unchanged. However, when there is stronger contamination, the dependence on initialization becomes more pronounced. (Table 2)
>
> Similarly, an ablation study comparing the variability of the distance between the optimized MDS and an oracle summary across perturbed initialization increases as contamination grows (Table 3). Overall, this suggests that there is a more challenging optimization landscape under more severe misspecification. Thus, in this setting, considering optimization techniques such as multiple initializations would be useful. We will include further experiments and discussions to clarify this issue more clearly in future revisions.
>
> | $\varepsilon$ | $\mathbf{s}_0$ | $\mathbf{s}_0$ (Gaussian perturbed) | Zero vector |
> | --- | --- | --- | --- |
> | 0.1 | 0.619 (0.039) | 0.620 (0.036) | 0.618 (0.039) |
> | 0.3 | 0.668 (0.042) | 0.671 (0.042) | 0.630 (0.039) |
> | 0.5 | 0.790 (0.041) | 0.807 (0.043) | 0.669 (0.039) |
>
> Table 2: RMSE against true parameter in OUP model with increasing contamination level for different initializations (standard error in parentheses)
>
> | $\varepsilon$ | Std. across perturbation |
> | --- | --- |
> | 0.1 | 0.013 |
> | 0.3 | 0.323 |
> | 0.5 | 0.460 |
>
> Table 3: Standard deviation, across Gaussian-perturbed random initializations, of the distance between the optimized MDS and the oracle summary, in the OUP model with increasing contamination level
>
>
> 3. We used the standard median heuristic (Gretton et al., 2008) as the choice of bandwidth. With the OUP model, we ran ablation studies scaling the median-heuristic bandwidth by a scaling factor of $k$. We find that the bandwidth choice is relatively stable around the median heuristic, however, for extreme under or over smoothing, the posterior accuracy substantially degrades (Table 4). This is in line with the expected behavior for the MMD objective, as there is a robustness tradeoff in the choice of bandwidth (Briol et al., 2019). While we find that the median heuristic is a good default choice, a more sophisticated tuning strategy may involve using a held-out validation set, and choosing the bandwidth based on a suitable downstream metric. We will add this ablation and further investigation of the choice of kernel bandwidth in future revisions.
>
> | $k$ | RMSE |
> | ---: | ---: |
> | 0.25  | 0.833 (0.051) |
> | 0.5 | 0.742 (0.042) |
> | 1.0 | 0.627 (0.040) |
> | 4.0 | 0.625 (0.040) |
> | 16.0 | 0.847 (0.046) |
>
> Table 4: RMSE against true parameter in OUP model with different scaling factor $k$ of the median heuristic, $\varepsilon = 0.2$
>
> > Minor issues
>
> Thank you for the feedback, and we will check the references carefully and fix the presentation in future revisions.

---

> > ### Author Rebuttal · Reviewer_PqPG · 2026-04-01
> >
> > I thank the authors for their responses, and appreciate the new discussion on misspecification and the ablation. The proposed changes make the paper more complete, and I have no follow-up questions. Thank you once again for your engagement and taking the time to do the extra experiments!

---

### Official Review · Reviewer_gYZL · 2026-03-16

**Soundness:** 2
**Presentation:** 3
**Significance:** 2
**Originality:** 3
**Overall Recommendation:** 5
**Confidence:** 3

**Summary:**

This work proposes a robust SBI framework that maintains the amortization properties of NPE but adapts observed summary statistics at test time to reduce misspecification. The main contribution is the proposal of minimum-distance summaries, a learnable function that maps simulated or observed data to an optimal summary statistic relative to a divergence. MMD is chosen here for its robustness properties and computational efficiency using random Fourier feature approximation. The authors demonstrate the robustness and consistency guarantees of the proposed approach, and apply it on multiple benchmark tasks while comparing to existing standard and robustified NPE algorithms.

**Compliance With Llm Reviewing Policy:**

Affirmed.

**Final Justification:**

The rebuttal fully addressed my concerns---this is an interesting and technically sound paper.

**Key Questions For Authors:**

1. Could you clarify what s, mu, phi, and z for the training set and test observations are?
2. If I understand correctly, the decoder compresses 100 iid observations into a single summary in the experiments. How does the method and results scale with the number of observations?
3. How would the proposed method be integrated into an end-to-end learned embedding network, and for sequential NPE?
4. How fast is optimization at test time (steps 9-13 in Algorithm 1)? Is it reasonable to still consider this amortized?

**Limitations:**

The paper adequately discusses technical limitations and societal risk.

**Strengths And Weaknesses:**

Soundness:
The approach and theorems appear to be sound though I did not carefully check the math. The evaluations are comprehensive, though only on artificial tasks under a specific form of misspecification, i.e., outliers. Parameter retrieval accuracy appears to be good in most cases but existing baselines are still quite competitive (though the rigorous evaluation is appreciated).

Presentation:
Overall I felt that the paper is well written, concise, and the visualizations are mindfully designed. I found some of the math hard to follow due to the notations sprinkled throughout the text (e.g., mu, phi, and z), and may have added to my confusion. Perhaps consider expanding the definition table.

Significance:
This work tackles a fundamental challenge in SBI, namely developing algorithms that are robust to model misspecification. The approach is interesting, but as mentioned above, is limited in its application to synthetic tasks under one specific form of misspecification, thus reducing its applicability. Maintaining (inference-)amortization is valuable though, and the paper appropriately contextualizes its contribution while highlighting its potential benefits.

Originality:
This work extends on a literature of SBI / NPE approaches using MMD as a robust divergence, and builds on that in an interesting way while providing a thorough overview of existing approaches.

---

> ### Author Rebuttal · Authors · 2026-03-31
>
> Dear Reviewer gYZL:
>
> We thank the reviewer for their thoughtful feedback.
>
> # Weaknesses
>
> We agree that the scope of misspecification should be clarified, and we will revise our manuscript to make this distinction clear. Our experiments focused primarily on the Huber model of contamination, where the observed distribution is $\mathbb{P} = (1-\varepsilon) \mathbb{P}_0 + \varepsilon \mathbb{Q}$, with $\mathbb{P}_0$ as the true distribution and $\mathbb{Q}$ as an arbitrary contaminating distribution, as our theoretical robustness guarantee is formulated in that setting. However, we note that the contaminating distribution $\mathbb{Q}$ can be arbitrary, and so can represent structured forms of contamination. This contamination model is common in robust SBI (Huang et al., 2023; Dellaporta et al., 2022), and while our guarantee is local in $\varepsilon$, empirically, we find that the MDS procedure performs well for a range of $\varepsilon$ values.
>
>
> In the SIR experiment, the type of misspecification is a form of **structured misspecification** where weekend counts are systematically underreported. We extend this by increasing the misspecification $\varepsilon$, where at $\varepsilon = 1$, all observed trajectories are distorted by the structured misspecification. Even in this regime, the MDS procedure continues to provide robustness benefits over the NPE baseline (Table 1), providing empirical evidence that MDS is effective not only for outlier contamination, but also for **structured contamination mechanisms**.
>
> We will revise the paper to include further experiments with structured misspecification to better evaluate our MDS procedure in this setting.
>
> | $\varepsilon$ | NPE | NPE-MDS |
> |---|---:|---:|
> | 0.6 | 0.0502 (0.0049) | **0.0271** (0.0042) |
> | 0.8 | 0.0621 (0.0052) | **0.0455** (0.0057) |
> | 1.0 | 0.0730 (0.0053) | **0.0593** (0.0082) |
>
> Table 1: RMSE against the true parameter in the SIR model with increasing misspecification fraction (lower is better, standard error in parentheses)
>
> # Questions
>
> 1. $\mathbf{s}$ is the summary statistic of a dataset, either the training or test dataset. $\phi$ is the kernel feature map for MMD, $z(\mathbf{x})$ is the random Fourier feature approximation of the kernel feature map. $\mu(\mathbf{s}) = \mathbb{E}[\phi(\mathbf{x}) \mid \mathbf{s}]$ represents the summary-conditional mean embedding, which is approximated with our decoder model using training datasets and used to obtain the MDS at test-time. We will revise the definition table in future revisions to make the definitions clearer.
>
> 2. To clarify, the $N$ observations are compressed by the summary statistic function. The decoder model instead maps a summary statistic to a conditional mean embedding. Computationally, the only dependency on the number of observations $N$ is through the calculation of the empirical mean embeddings $\frac{1}{N} \sum_{n=1}^N z(\mathbf{x}_n)$, which is linear in $N$, subsequent test-time optimization is over the summary statistic and is independent of $N$. Empirically, our sample-size ablation (Appendix D.1.1 of the manuscript) shows that the robustness gains generally improve as $N$ increases, which is consistent with the mean embeddings becoming less noisy.
>
> 3. Yes, the MDS procedure can be extended to learned summary networks and SNPE, as it is decoupled from the training process of the NPE. After the initial joint training of the encoder and NPE, we can freeze the learned encoder and treat it as a fixed summary statistic function to be used with our MDS procedure, this is already the setting in our Gaussian experiment. We further extend the OUP model experiment using a learned neural summary, and observe that MDS is able to provide similar robustness benefits (Table 2). For SNPE, a natural extension is to apply the MDS after the final round of the SNPE training, and use the simulated datasets pooled across sequential training. However, because the proposal distribution changes across rounds, the decoder training could be extended to account for this difference. We will provide further discussions of this in future revisions.
>
> | $\varepsilon$ | NPE | NPE-MDS |
> |---|---:|---:|
> | 0.0 | **0.640** (0.039) | 0.642 (0.039) |
> | 0.1 | 1.072 (0.046) | **0.642** (0.039) |
> | 0.3 | 1.159 (0.034) | **0.661** (0.039) |
> | 0.5 | 1.429 (0.043) | **0.801** (0.04) |
>
> Table 2: RMSE against the true parameter in the OUP model with learned neural embedding
>
> 4. Test-time MDS adaptation is lightweight, across experiments L-BFGS optimization typically converges in about 10 to 30 iterations, taking about 0.03s per query, which is about two orders of magnitude faster than the offline decoder model training (which takes ~6s). Test-time optimization is deterministic, and done on the typically low-dimensional summary space, without additional simulator calls. Thus, we view the MDS as preserving the amortization property of the inference engine, adding only minimal per query adaptation cost for robustness.

---

> > ### Author Rebuttal · Reviewer_gYZL · 2026-04-03
> >
> > I appreciate the clarifications which aided in my understanding. My concerns are fully resolved and I will raise my score to 5, recommending acceptance.

---

### Decision · Program_Chairs · 2026-04-30

**Decision:**

Accept (regular)

**Comment:**

This paper proposes a method to cope with model misspecification for simulation-based inference.  Reviewers unanimously recommended acceptance and stated that the authors fully resolved their questions in their rebuttal.